# Combining Fe nanoparticles and pyrrole-type Fe-N$_4$ sites on less-oxygenated carbon supports for electrochemical CO$_2$ reduction

Cai Wang[1], Xiaoyu Wang[1], Houan Ren[1], Yilin Zhang[1], Xiaomei Zhou[1], Jing Wang[1], Qingxin Guan[1], Yuping Liu ®[1] & Wei Li[1] ✉

A great challenge for electrochemical CO$_2$ reduction is to improve energy efficiency, which requires reducing overpotential while increasing product Faraday efficiency. Here, we designedly synthesize a hybrid electrocatalyst consisting of Fe nanoparticles, pyrrole-type Fe-N$_4$ sites and less-oxygenated carbon supports, which exhibits a remarkable CO Faraday efficiency above 99% at an ultralow overpotential of 21 mV, reaching the highest cathode energy efficiency of 97.1% to date. The catalyst also can afford a CO selectivity nearly 100% with a high cathode energy efficiency (>90%) at least 100 h. The combined results of control experiments, in situ characterizations and theoretical calculations demonstrate that introducing Fe nanoparticles can reduce the overpotential by accelerating the proton transfer from CO$_2$ to *COOH and lowering the free energy for *COOH formation, constructing pyrrole-type Fe-N$_4$ sites and limiting oxygen species on carbon supports can increase CO Faraday efficiency through inhibiting the H$_2$ evolution, thus achieving energy-efficient electrochemical CO$_2$ reduction to CO.

Electrochemical CO$_2$ reduction (ECR) as a sustainable approach has attracted much attention on the CO$_2$ transformations and carbon-neutral economic cycle[1,2]. CO, which is an essential feedstock for the production of muti-carbon chemicals, is one of the most valuable ECR products[3,4]. Though numerous works have been dedicated to the developing of catalysts for CO$_2$-to-CO electrolysis and the reported parameters about activity and selectivity have achieved industrial level[5–9], rare electrocatalysts can work with high energy efficiency. As a result, there is an urgent demand to excavate and screen an advanced electrocatalyst for energy-efficient ECR, with the core challenge of reducing overpotential while keeping high Faraday efficiency.

Currently, noble metal electrocatalysts, mainly Au and Ag, exhibit a superior performance to produce CO at low potential[10–12]. For instance, the Sun group[11] reported that ultrathin Au nanowires present an ultralow onset potential of −0.2 V and a high CO Faradaic efficiency (FE) of 94% at −0.35 V. Nevertheless, the relatively scarcity and high cost of noble metal catalysts prompted researchers to focus on exploring non-precious metal catalysts. Recent studies have shown that cheap transition metal and nitrogen codoped carbon (M-N-C) catalysts with single M-N$_x$ sites possess remarkable performance in ECR to CO[13–17], of which Fe-N-C are the great prospective catalysts in achieving high FE$_{CO}$ at low overpotential. Typically, Gu et al.[18] developed a Fe$^{3+}$-N-C catalyst that could produce CO at a quite low overpotential of 80 mV, and Zhang et al.[19] reported a Fe-N-C catalyst namely FeN$_5$ reached a high FE$_{CO}$ of 97.0% at a low overpotential of 0.35 V. Despite these breakthroughs, for Fe-N-C electrocatalysts, there is still a large room for advance in improving FE$_{CO}$ and reducing overpotential.

Regulating the coordination structure of the central metal atom can improve the selectivity and activity of M-N-C catalysts on ECR reaction, which often optimizes the formation/desorption of ECR intermediates by altering the electronic and geometric structures of the active sites. Thereinto, adjusting the type of coordination N atoms is an effective strategy to tune the selectivity of reduction product[18,20].

[1]State Key Laboratory of Elemento-Organic Chemistry, Key Laboratory of Advanced Energy Materials Chemistry (Ministry of Education), College of Chemistry, Nankai University, Tianjin 300071, China. ✉e-mail: weili@nankai.edu.cn

However, there are still large challenges in the controllable preparation of M-N-C catalysts with well-defined coordination N type to date. Furthermore, metal nanoparticles (NPs) are generally avoided during the preparation of M-N-C or removed through subsequent pickling treatment due to the negative impact on ECR performance[14,16,18,19], while other recent works demonstrated that anchoring metal NPs on M-N-C can boost the ECR performance by accelerating the proton transfer[21,22]. In fact, metal NPs are able to theoretically stabilize intermediates in the reaction process due to its abundant metal sites with high binding energy, so introducing Fe NPs on Fe-N-C may be a hopeful strategy to reduce the overpotential on ECR reaction, even though the relevant investigations are rarely reported. Another noteworthy aspect is that the oxygen species containing various oxygen-containing groups on carbon supports are impossible to overlooked, they have been proved to affect the catalytic performance of M-N-C[23]. The above analysis inspires us to further precisely design and modify Fe-N-C electrocatalysts for achieve highly energy-efficient ECR.

In this work, a modified Fe-N-C electrocatalyst, Fe NPs and pyrrole-type Fe-N$_4$ sites immobilized on less-oxygenated carbon matrix (Fe-poN-C/Fe), was elaborately prepared. For comparison, oxygenated carbon matrix supported pyridine-type Fe-N$_4$ sites (Fe-pdN-C/(O)) and pyrrole-type Fe-N$_4$ sites (Fe-poN-C/(O)) were also prepared. The Fe-poN-C/Fe catalyst delivers a high FE$_{CO}$ of 99.7% at a low overpotential of 0.24 V in an H-type cell, which also achieves an ultrahigh cathode energy efficiency (CEE) of 97.1% with nearly 100%

FE$_{CO}$ and a current density of −14.1 mA·cm$^{-2}$ at an ultralow overpotential of 21 mV in a flow cell, outperforming Fe-pdN-C, Fe-poN-C and almost all ECR catalysts as far as we know. Furthermore, stability tests over 100 h show that Fe-poN-C/Fe catalysts can produce CO with a high CEE (>90%) and nearly 100% CO selectivity at a current density over 40 mA · cm$^{-2}$. A series of experimental measurements and in situ characterization reveal that pyrrole-type Fe-N$_4$ sites and less-oxygenated carbon supports are beneficial to improve CO selectivity, and Fe NPs can reduce the overpotential through facilitating the proton transfer. DFT calculations corroborate the above findings and further give a deep theoretical understanding.

## Results and discussion

### Catalyst preparation and characterization

As shown in Fig. 1a, three different Fe-N-C catalysts were prepared through selecting different nitrogen precursor and adjusting pyrolysis condition (detailed prepared procedures as shown in Supporting Information). Thereinto, Fe-pdN-C(O) and Fe-poN-C(O) were obtained after pyrolysis at 600 °C in Ar atmosphere using Fe(III)-phenanthroline (Fe-phen) complex and Fe(III) tetraphenylporphyrin (FeTpp) as precursors, respectively. For Fe-poN-C/Fe, the FeTpp was employing as a precursor and the pyrolysis condition was optimized to 700 °C and H$_2$/Ar(5:95) atmosphere. Altering the type of coordination N in the precursor may lead to a different coordination structure in Fe-N-C catalysts. The high temperature and H$_2$ atmosphere are crucial for the

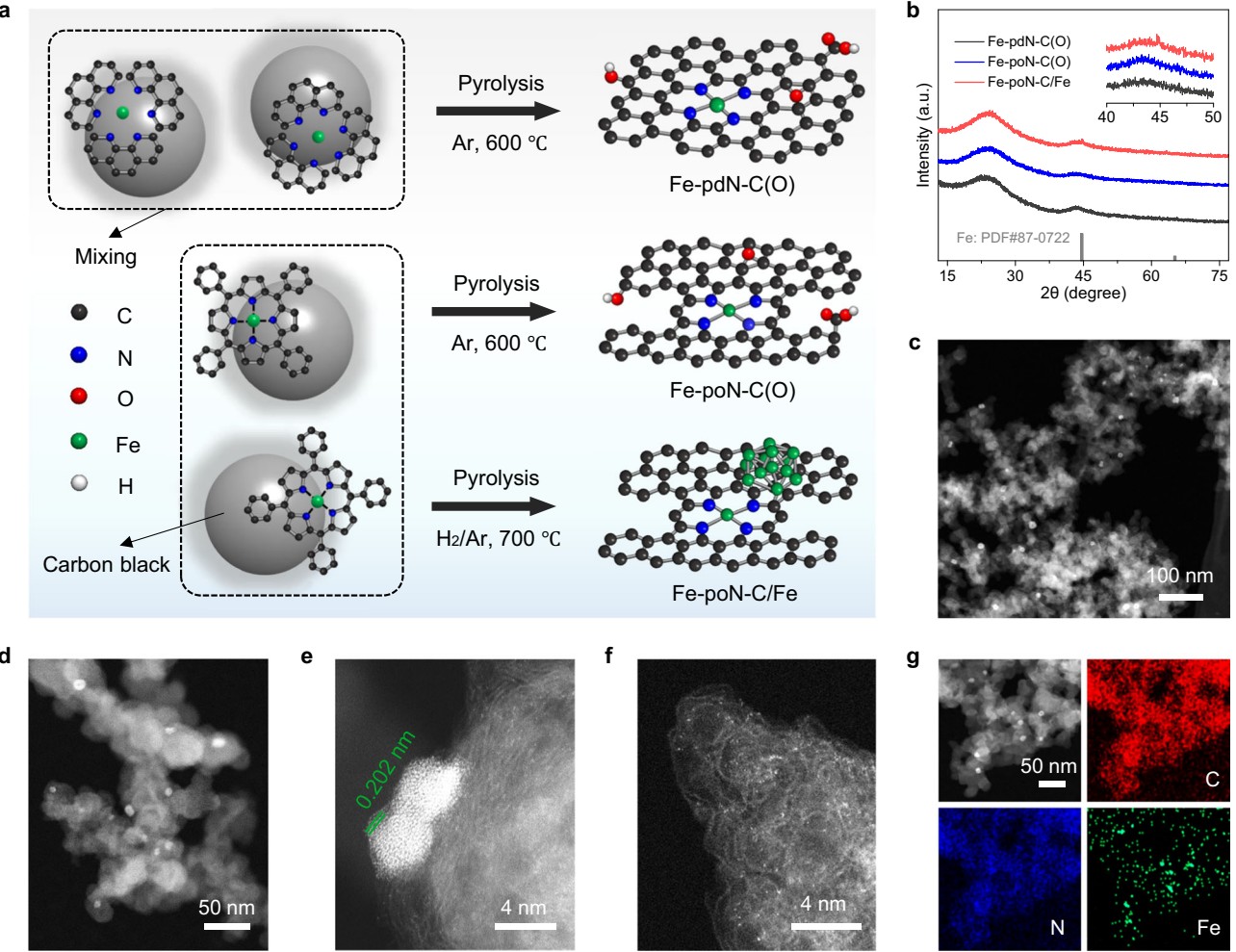

**Fig. 1 | Preparation and characterization of catalyst. a** Schematic illustration for the synthesis of Fe-pdN-C(O), Fe-poN-C(O) and Fe-poN-C/Fe. **b** XRD patterns of Fe-pdN-C(O), Fe-poN-C(O) and Fe-poN-C/Fe. (**c**) Large-field view, (**d**) magnified view of HAADF-TEM, (**e**, **f**) aberration-corrected HAADF-STEM, and (**g**) EDS mapping of Fe-poN-C/Fe.

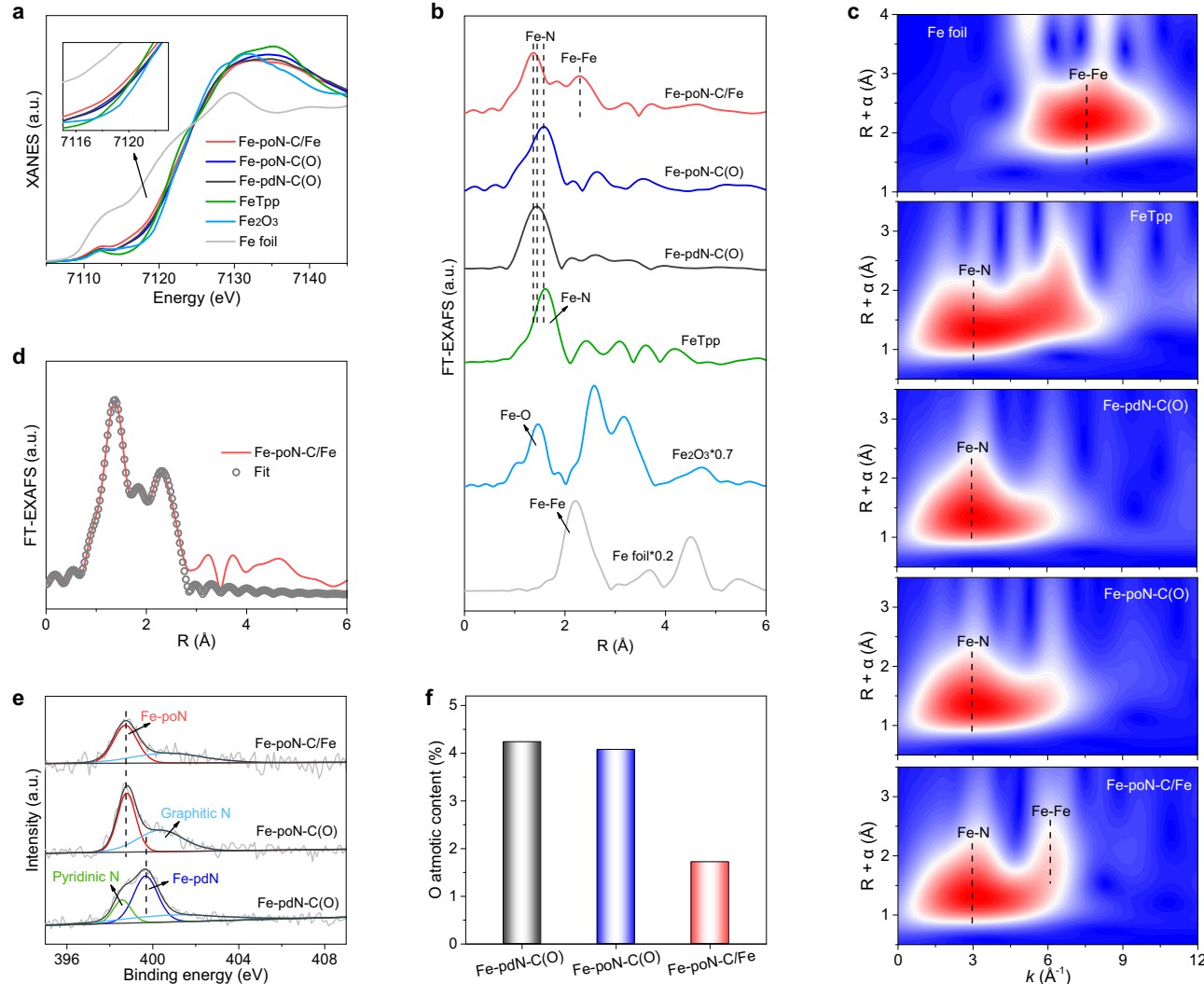

**Fig. 2 | X-ray absorption spectroscopy and X-ray photoelectron spectroscopy analyses.** (**a**) XANES and (**b**) FT-EXAFS spectra at the Fe K-edge of Fe-pdN-C(O), Fe-poN-C(O) and Fe-poN-C/Fe with the reference samples. (**c**) WT-EXAFS contour plots of Fe foil, FeTpp, Fe-pdN-C(O), Fe-poN-C(O) and Fe-poN-C/Fe. (**d**) The fitting curve of FT-EXAFS for Fe-poN-C/Fe. (**e**) N 1 s spectra and (**f**) O atomic content of Fe-pdN-C(O), Fe-poN-C(O) and Fe-poN-C/Fe.

introduction of Fe NPs and the decrease of oxygen species on carbon supports[24,25].

The X-ray diffraction (XRD) patterns (Fig. 1b) of three catalysts exhibit two similar peaks for amorphous carbon at 20-30° and 40-50°[14], besides, Fe-poN-C/Fe shows a small peak for Fe metal phase. No peaks of any crystalline species of Fe were observed in the XRD of Fe-pdN-C(O) and Fe-poN-C(O), indicating that Fe may exist in the form of single atom or small clusters. Likewise, the HAADF-STEM of Fe-poN-C/Fe reveals some obvious Fe nanoparticles (NPs) at a size around 3-8 nm (Fig. 1c), while no NPs were found in the HAADF-STEM of Fe-pdN-C(O) and Fe-poN-C(O) (Fig. S1a and S2a). Notably, a magnified view of HAADF-TEM (Fig. 1d) and aberration-corrected HAADF-TEM (Fig. 1e and S3) present that the Fe NPs with a lattice distance of 0.202 nm are primarily located at the surface of the carbon support, rather than the carbon-shell-encapsulated structure. Furthermore, numerous bright dots assigned to Fe single atoms were observed on the carbon supports (Fig. 1e, f). The HAADF-STEM image with EDS mapping results (Fig. 1g) displays the uniform distribution of C, N, Fe and segregated Fe NPs in the Fe-poN-C/Fe. Such a local structure in Fe-poN-C/Fe may result in a special interaction between Fe NPs and atomic-level Fe. Only Fe single atoms were observed in Fe-pdN-C(O) and Fe-poN-C(O) (Figs. S1, 2). Inductively coupled plasma-optical emission spectrometer

measurements show the Fe contents in Fe-pdN-C(O), Fe-poN-C(O) and Fe-poN-C/Fe are 1.43 wt%, 1.20 wt% and 1.14 wt%, respectively. Similar Fe contents in three catalysts indicate that the performance difference is related to the local coordination structure rather than the metal contents, as discussed below.

The fine structure of Fe species in the prepared samples was identified by synchrotron X-ray absorption spectroscopy (XAS) analysis. The Fe valence states in Fe-pdN-C(O), Fe-poN-C(O) and Fe-poN-C/Fe were detected through the Fe K-edge X-ray absorption near-edge structure (XANES) spectra. As shown in Fig. 2a, though the near-edge absorption of Fe-pdN-C(O) and Fe-poN-C(O) lies between those of Fe foil and $Fe_2O_3$, they are closer to that of FeTpp, indicating that the Fe atom in Fe-pdN-C(O) and Fe-poN-C(O) with a positive oxidation state nearly +3. However, an apparent shift of the near-edge absorption toward lower energy was noted over Fe-poN-C/Fe, implying a lower oxidation state of Fe species in Fe-poN-C/Fe. This phenomenon can be ascribed to the coexistence of Fe NPs and single Fe atom in Fe-poN-C/Fe, which is in line with the XRD and HAADF-STEM results. Next, the local coordination structure was further explored by the Fourier transform extended X-ray absorption fine structure (FT-EXAFS) spectra. The FT-EXAFS spectra of the Fe K-edge in Fig. 2b show that the primary peaks at 1.45 Å for Fe-pdN-C(O) and 1.57 Å for Fe-poN-C(O) can

be attributed to the presence of Fe-N coordination bond. No obvious peaks about the Fe-Fe bond at ~2.2 Å were observed, revealing that atomical dispersion of Fe in Fe-pdN-C(O) and Fe-poN-C(O). The slight shift of the Fe-N peak between the Fe-pdN-C(O) and Fe-poN-C(O) may due to the different Fe-N coordination structures. Considering the different type of coordination N in precursor and the position of the Fe-N peak in the Fe-poN-C(O) is close to that in FeTpp, we infer that Fe-poN-C(O) mainly contains the pyrrole-type Fe-N structure (Fe-poN) while the Fe-pdN-C(O) may dominate the pyridine-type Fe-N (Fe-pdN) structure. The FT-EXAFS spectrum of Fe K-edge for Fe-poN-C/Fe displays two main peaks at around 1.37 and 2.28 Å, which can be attributed to the scattering path of Fe-N and Fe-Fe, further indicating the co-existence of single Fe-$N_x$ sites and Fe NPs in Fe-poN-C/Fe. A small peak at 1.84 Å can be attributed to the Fe-C bond that exists between Fe NPs and carbon supports. The Fe-N bond length in Fe-poN-C/Fe is shorter than that in Fe-poN-C(O), which may derive from the strong interaction between Fe NPs and Fe-$N_x$ groups rather than the difference of Fe-N coordination structure[26,27]. The contour plots from EXAFS wavelet transform (WT) analysis are shown in Fig. 2c, only one intensity maximum at about 3 Å$^{-1}$ were observed in Fe-pdN-C(O) and Fe-poN-C(O), which can be assigned to Fe-N bond. The contour plot of Fe-poN-C/Fe shows an intensity maximum at approximately 3 Å$^{-1}$ and a second intensity maximum at approximately 6.1 Å$^{-1}$ that were ascribed to the Fe-N and Fe-Fe bond, respectively. The negative shift of the intensity maximum corresponding to the Fe-Fe bond of Fe foil (-7.6 Å$^{-1}$) may because of the coordination number difference between Fe NPs and bulk Fe[28]. The FT-EXAFS curve of Fe-poN-C/Fe was fitted well (Fig. 2d) and the coordination numbers of Fe-N coordination structure and Fe-Fe species were approximately 3.9 and 1.3 (Table S1), respectively. Meanwhile, the coordination numbers of the Fe-N coordination structure in Fe-pdN-C(O) and Fe-poN-C(O) were about 4.1 and 4.2 (Fig. S4 and Table S1), respectively. These results support the well-defined Fe-$N_4$ unit exist in the three samples.

Next, the X-ray photoelectron spectroscopy (XPS) was performed to ascertain the chemical composition (Fig. S5 and Table S2). As shown in the high-resolution N 1 s spectra (Fig. 2e), the primary peaks of the three catalysts have a prominent difference at the energy position. Previous reports manifested that the type of coordination N species in the M-N structure was pyridinic-N or pyrrolic-N, and different coordination N would lead to differences in the energy position[15,29,30]. In our work, the main peak at around 398.7 eV from Fe-poN-C/Fe and Fe-poN-C(O) can be ascribed to the Fe-poN structure, and the main peak at around 399.7 eV from Fe-pdN-C(O) can be ascribed to the Fe-pdN structure, which agrees with the XAS analysis and previous studies[18,31]. Moreover, the comparison of the N 1 s spectra of Fe-pdN-C(O) and Fe-poN-C(O) with that of Fe-phen complex and FeTpp further confirmed that the types of coordination N in Fe-pdN-C(O) and Fe-poN-C(O) are mainly pyridine N and pyrrole N, respectively (Fig. S6)[18]. The other inconspicuous peaks at around 398.9 eV in Fe-pdN-C(O) and 400.6 eV in three catalysts can be attributed to the low contents of pyridinic-N and graphitic-N, respectively. Although oxygen-containing species are not deliberately introduced during preparation, the oxygen signals were detected in the full XPS spectra for the three samples (Figure. S5). This phenomenon may be caused by the oxygen-containing species of the carbon black carrier itself or the influence of air in the evaporation process, which has occurred in other reported M-N-C materials[32–35]. Fig. 2f and Table S1 show that the oxygen contents of Fe-pdN-C(O) and Fe-poN-C(O) are higher than that of Fe-poN-C/Fe, indicating more oxygen species on the carbon supports of Fe-pdN-C(O) and Fe-poN-C(O).

## Electrochemical CO₂ reduction performance

The ECR performance on prepared catalysts was firstly assessed in a three-electrode H-cell containing $CO_2$-saturated 0.5 M $KHCO_3$ electrolyte. Linear sweep voltammetry (LSV) measurements reveal that Fe-poN-C/Fe possesses a higher geometric current density ($j$) and lower onset overpotential than that of Fe-pdN-C(O) and Fe-poN-C(O) (Fig. 3a), indicating the potentially remarkable catalytic performance of Fe-poN-C/Fe on ECR. Potentiostatic electrolysis was performed to determine reduction products with the assistance of online gas chromatography (GC) and $^1$H NMR spectroscopy. CO and $H_2$ are the only gaseous products, and no liquid products were detected. As shown in Fig. 3b, the CO selectivity of Fe-poN-C/Fe is obviously superior to the other two catalysts, it maintains over 95% $FE_{CO}$ at a wide potential range (−0.3 to −0.7 V) with the maximum $FE_{CO}$ of 99.7% at −0.35 V. Also, Fe-poN-C/Fe offers a high $FE_{CO}$ of 96.6% at a low potential of −0.3 V, far exceeding that of Fe-pdN-C(O) (38.3%) and Fe-poN-C(O) (71.3%). Meanwhile, an ultralow $FE_{H_2}$ nearly 0% from −0.3 V to −0.55 V is achieved on Fe-poN-C/Fe (Fig. S7), displaying its superior capability of inhibiting hydrogen evolution reaction (HER). Impressively, Fe-poN-C/Fe can realize approximately 100% $FE_{CO}$ at a quite low overpotential, outperforms or rivals previously reported $CO_2$-to-CO electrocatalysts, such as Fe-N-C catalysts, noble metal catalysts and other state-of-the-art catalysts (Fig. 3c and Table S3). This makes Fe-poN-C/Fe a promising catalyst for energy-efficient ECR.

The ECR activity of three catalysts was evaluated by calculating the CO current density ($j_{CO}$) and turnover frequency (TOF). The $j_{CO}$ of Fe-poN-C/Fe (−15.1 mA·cm$^{-2}$) at −0.65 V is much larger than that of Fe-pdN-C(O) (−1.5 mA·cm$^{-2}$) and Fe-poN-C(O) (−4.1 mA·cm$^{-2}$) (Fig. 3d). The calculated CO TOF of Fe-poN-C/Fe is also higher than that of Fe-pdN-C(O) and Fe-poN-C(O) under the applied potential range, which reaches a high value of 2890.5 h$^{-1}$ at a potential of −0.75 V (Fig. S8). These results suggest the remarkable activity of Fe-poN-C/Fe for CO production. The electrochemically active surface area (ECSA) of these catalysts is assessed by measuring the double-layer capacitance ($C_{dl}$) to explore the origin of the improved ECR activity on Fe-poN-C/Fe (Fig. S9). It can be seen from Fig. 3e that the order of three catalysts on $C_{dl}$ value is inconsistent to that on tested activity, the $C_{dl}$ value of Fe-poN-C/Fe (21.4 mF·cm$^{-2}$) in between Fe-pdN-C(O) (22.1 mF·cm$^{-2}$) and Fe-poN-C(O) (19.7 mF·cm$^{-2}$), demonstrating the enhanced ECR activity of Fe-poN-C/Fe is mainly ascribed to the intrinsic activity rather than surface area effect. Moreover, for gaining insight into ECR kinetic, the Tafel analysis and electrochemical impedance spectroscopy (EIS) tests were performed. The Tafel slope of Fe-poN-C/Fe is 79 mV·dec$^{-1}$, lower than those of Fe-pdN-C(O) (95 mV·dec$^{-1}$) and Fe-poN-C(O) (88 mV·dec$^{-1}$) (Fig. S10), indicating the CO production on Fe-poN-C/Fe proceed with a faster kinetics as compared with other two catalysts. The Nyquist plots show that Fe-poN-C/Fe possesses a lower charge transfer resistance as compared with Fe-pdN-C(O) and Fe-poN-C(O) (Figure. S11), indicating the significantly fast charge-transfer process and improved electronic conductivity after introducing Fe NPs, eventually resulting in an enhanced activity on ECR.

To identify the active sites responsible for the ECR performance, several control experiments were carried out. Firstly, two kinds of N-doped carbon (namely $N_1$-C and $N_2$-C) were prepared (Fig. S12a, b) and their ECR performance were evaluated. It can be seen from Figure. S12c, d and Fig. 3b, d that the $FE_{CO}$ and $j_{CO}$ of $N_1$-C and $N_2$-C are both significantly lower than that of Fe-pdN-C(O), Fe-poN-C(O) and Fe-poN-C/Fe, demonstrating the Fe species primarily serve as the active centers. Next, SCN$^-$ poisoning experiments were performed to confirm the role of Fe-$N_4$ sites because SCN$^-$ can poison Fe-N structure in catalyzing ECR[36,37]. After adding SCN$^-$, the $FE_{CO}$ and $j_{CO}$ of Fe-poN-C(O) and Fe-poN-C/Fe display obvious drop (Figure. S13, 14), indicating the critical role of isolated Fe-$N_4$ sites for the remarkable ECR performance of Fe-poN-C/Fe. The effect of Fe NPs and oxygen species on the ECR performance was further studied by control experiments. Firstly, Fe-poN-C(O) catalyst was treated with $H_2O_2$/$H_2SO_4$ solutions (0.5 M $H_2SO_4$ containing certain 30% $H_2O_2$) under 80 °C for 24 h to increase the content of oxygen species. XPS results exhibit that more oxygen species have been introduced on Fe-poN-C(O) after $H_2O_2$/$H_2SO_4$

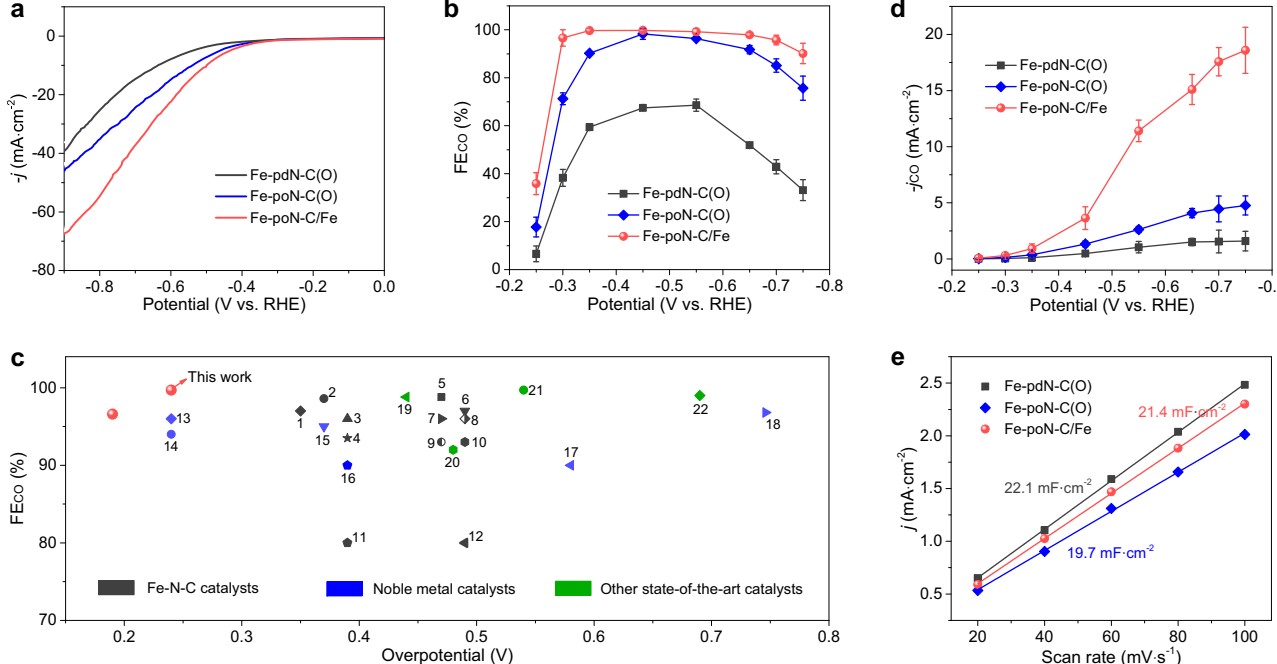

**Fig. 3 | Electrochemical CO$_2$ reduction performance in H-cell. a** LSV curves of Fe-pdN-C(O), Fe-poN-C(O) and Fe-poN-C/Fe. **b** FE$_{CO}$ at various applied potentials of Fe-pdN-C(O), Fe-poN-C(O) and Fe-poN-C/Fe. **c** ECR performance of Fe-poN-C/Fe as compared with that of typical electrocatalysts (see Table S3). **d** $j_{CO}$ at various applied potentials and (**e**) the double-layer capacitance of Fe-pdN-C(O), Fe-poN-C(O) and Fe-poN-C/Fe. All measurements in H-cell are performed in CO$_2$-saturated 0.5 M KHCO$_3$ solution (pH 7.3, resistance is 7.35 Ω), and the catalyst mass loading is 0.5 mg·cm$^{-2}$. Error bars in (**b**) and (**d**) represent the standard deviation of three independent measurements.

treatment (Fig. S15a). The FE$_{CO}$ and j$_{CO}$ of the treated Fe-poN-C(O) (named as Fe-poN-C(O)-(H$_2$O$_2$/H$_2$SO$_4$)) decreased significantly as compared with Fe-poN-C(O) (Fig. S15b, c), indicating the oxygen species on carbon supports are unfavorable for promoting ECR. Second, the Fe-poN-C/Fe catalyst was treated with H$_2$SO$_4$ solutions (0.5 M H$_2$SO$_4$) under 80 °C for 24 h to remove the Fe NPs. The HAADF-STEM images of the treated Fe-poN-C/Fe (named as Fe-poN-C/Fe-(H$_2$SO$_4$)) show that the Fe NPs are almost all removed and single-atom Fe sites remained (Fig. S16a, b). Performance tests exhibits the FE$_{CO}$ at low potentials and j$_{CO}$ decreased after the removal of Fe NPs (Fig. S16c, d), which indicates that the existence of Fe NPs is benefit for reducing overpotential on ECR. Furthermore, the HAADF-STEM images and XPS results shown in Fig. S17a–c reveal the Fe NPs in Fe-poN-C/Fe are removed while more oxygen species are introduced after H$_2$O$_2$/H$_2$SO$_4$ treatment. An apparent decline in the FE$_{CO}$ and $j_{CO}$ of Fe-poN-C/Fe-(H$_2$SO$_4$/H$_2$O$_2$) was observed (Fig. S17d, e), combined with the above two control experiments, which means that the Fe NPs possess a positive role in reducing overpotential and oxygen species exhibits a negative role in improving CO selectivity.

According to the aforesaid discussion, the superior performance of Fe-poN-C/Fe at low overpotential can be ascribed to the following three points: (i) the construction of pyrrole-type Fe-N$_4$ sites is essential to deliver high CO selectivity, which can be seen from the performance comparison between Fe-pdN-C(O) and Fe-poN-C(O); (ii) the introduction of Fe NPs is the key to reduce the overpotential; (iii) the removal of oxygen species on carbon supports is conducive to further improving CO selectivity.

Considering the excellent CO$_2$ reduction selectivity of Fe-poN-C/Fe in a traditional H-type cell, its high-current activity was further evaluated in a flow cell device equipped with gas diffusion electrode as well as the electrolyte was 1 M KOH (Fig. S18). LSV curves in a flow cell (Fig. 4a) show a significantly higher current density of Fe-poN-C/Fe as compared with that of Fe-pdN-C(O) and Fe-poN-C(O). Impressively, Fe-poN-C/Fe exhibits an ultralow onset potential of −0.12 V with a FE$_{CO}$ of

83.2% and the high FE$_{CO}$ over 99% could reach at −0.15 V (Fig. 4b). When the applied potential increased from −0.15 to −0.7 V, the FE$_{CO}$ of Fe-poN-C/Fe has remained above 95%, and a high current density of 154.3 mA·cm$^{-2}$ at −0.7 V was achieved. Especially, Fe-poN-C/Fe possesses a remarkable FE$_{CO}$ over 99% at an overpotential of 41 mV without *iR* correction or 21 mV with *iR* correction, outperforming practically all state-of-the-art electrocatalysts for ECR to CO in a flow cell (Fig. 4c and Table S4). Energy efficiency is an essential factor to prove whether ECR could be industrialized. The cathode energy efficiency (CEE) for CO production on Fe-poN-C/Fe could achieve >80% under a wide potential window from −0.112 to −0.48 V with *iR* correction (Fig. S19a), and the maximum CEE of 97.1% with a current density of −14.1 mA·cm$^{-2}$ was obtained at an ultralow overpotential of 21 mV, this performance is optimal to date (Fig. 4d and Table S5). More importantly, the stability of Fe-poN-C/Fe is evaluated under an applied potential of −0.3 V (Fig. 4e and S19b), the 100 h tested results show that Fe-poN-C/Fe can deliver a stable FE$_{CO}$ almost 100 % with a current density above 40 mA·cm$^{-2}$, which also can maintain a high CEE over 90%. In addition, the single-atom Fe sites and Fe NPs are well preserved after the long-term testing (Fig. S20), further indicating the remarkable durability of Fe-poN-C/Fe for CO production.

**Mechanism investigation**

In situ Raman measurements using the custom Raman cell were adopted to monitor the evolution of catalyst surface adsorbates during ECR. The Raman spectra were recording in CO$_2$-saturated 0.5 M KHCO$_3$ under a constant potential of −0.35 V. The Raman spectra of Fe-poN-C(O) and Fe-poN-C/Fe during the electrolysis time of 0 to 100 s were shown in Fig. 5a and b, respectively. The peaks at around 1152, 1536 cm$^{-2}$ can be attributed to *CO$_2^-$, and the peaks at around 1017 cm$^{-2}$ kept invariable with the electrolysis time were assigned to the CO$_3^{2-}$ in the electrolyte[38–40]. The assignment of *CO$_2^-$ peaks on Fe-poN-C(O) appeared after applying potential and gradually strengthened with increasing electrolysis time, while no peaks corresponded to *CO$_2^-$

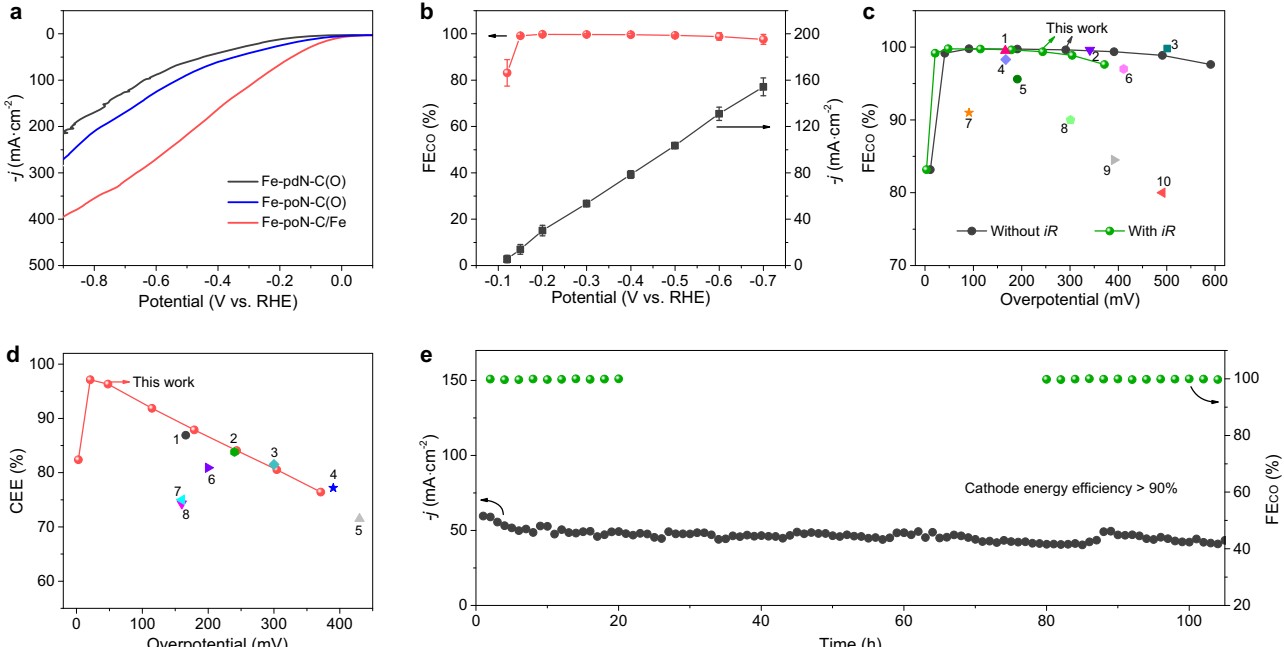

**Fig. 4 | Electrochemical CO₂ reduction performance in flow cell. a** LSV curves of Fe-pdN-C(O), Fe-poN-C(O) and Fe-poN-C/Fe. **b** FE$_{CO}$ and the corresponding *j* of Fe-poN-C/Fe at various potentials. (**c**) FE$_{CO}$ and (**d**) CEE of Fe-poN-C/Fe compared with those of most state-of-the-art catalysts (see Table S4,5). **e** Stability tests of Fe-poN-C/Fe under an applied potential of −0.3 V. All measurements in flow cell are performed in 1 M KOH solution (pH 13.7, resistance is 1.68 Ω), and the catalyst mass loading is 1 mg·cm⁻². Error bars in (**b**) represent the standard deviation of three independent measurements.

were observed on Fe-poN-C/Fe, implying that the *CO₂⁻ are likely to be rapidly protonated to the *COOH intermediates on the surface of Fe-poN-C/Fe. It is well known that the activation of H₂O plays a crucial role in the protonation process during ECR to CO[21]. Thus, the investigation on the kinetic isotope effect (KIE) over Fe-poN-C(O) and Fe-poN-C/Fe catalysts were conducted to further explore the effect of Fe NPs in H₂O activation progress during ECR[21,41,42]. As shown in Fig. 5c, the calculated KIE value of Fe-poN-C(O) is 1.41, while Fe-poN-C/Fe shows a much lower KIE value (1.14), indicating that the H₂O activation on Fe-poN-C/Fe catalysts is easy[21,41]. In other words, combined with the results of in situ Raman results, the introduction of Fe NPs in Fe-poN-C/Fe promotes the H₂O dissociation, and this is beneficial for accelerating proton transfer process and forming *COOH intermediates.

To deeply understand how the local structure, coordination N type, Fe NPs and the oxygen species on carbon, contributes to the performance of Fe-N-C catalysts on ECR to CO, DFT calculations were conducted. As shown in Fig. 5d, pyridine-type Fe-N₄ (Fe-pdN₄), pyrrole-type Fe-N₄ (Fe-poN₄) and Fe13 nanoclusters placed on pyrrole-type Fe-N₄ (Fe-poN₄/Fe13) were constructed as per the structure characterizations and other previous works[7,37,43,44], which were served as models for Fe-pdN-C(O), Fe-poN-C(O) and Fe-poN-C/Fe catalysts, respectively. We adopted a coupled proton-electron transfer associative ECR pathway that proceeds through *COOH and *CO (Fig. S21). The calculated free energy diagrams at −0.3 V are shown in Fig. 5e. Consistent with previous studies[19,34,45,46], the rate limiting step (RDS) on Fe-pdN₄ is the *CO desorption with a high free energy change of 1.16 eV. Although Fe-poN₄ exhibits lower free energy change (0.55 eV) for * CO desorption than that of Fe-pdN₄, its free energy change (0.16 eV) required for *COOH formation obviously increased. After introducing surface Fe NPs, Fe-poN₄/Fe13 greatly reduced the free energy gap for the *COOH formation (−0.86 eV) and *CO desorption (0.65 eV) as compared with Fe-poN₄ and Fe-pdN₄, thus achieving a superior activity and lower overpotential for CO production on Fe-poN-C/Fe catalyst. Furthermore, three catalyst models binding with the important *COOH intermediate were further investigated by differential charge

distribution. As shown in Fig. 5f–h, Fe-poN₄/Fe13 shows an obvious electron density accumulation between the active site and *COOH as compared with Fe-pdN₄ and Fe-poN₄, suggesting that the introduction of Fe NPs is favorable to stabilize the *COOH intermediate, thus boosting the CO₂ reduction process[47]. Besides, another model represented Fe-poN₄ encapsulated Fe NPs (Fe13@Fe-poN₄) was built, in which Fe-poN₄ as the active site (Fig. S22). Calculated results show that Fe13@Fe-poN₄ possesses a far higher free energy gap for *COOH formation (0.9 eV) than that of Fe-poN₄/Fe13, implying the importance of surface Fe NPs in promoting ECR activity.

To explore the difficulty degree of ECR reaction more actually, the activation energies for the three catalysts model along the CO₂ reduction pathway through a direct hydrogenation mechanism were assessed[48,49]. As shown in Fig. S23, the elementary reactions for *COOH formation and *CO formation are both exothermic, and the activation energies of the three catalysts model to form *COOH (0.15-0.33 eV) or *CO (0.24-0.63 eV) are relatively low because the activation energies of typical catalysts (Ag[48] and Ni-N-C[49]) used in ECR to form *COOH or *CO are much higher than 1 eV. This indicates that the hydrogenation steps of three catalysts during CO₂ reduction are easy on the premise of providing sufficient *H. In other words, the capacity of *H supply before CO₂ hydrogenation step is a key factor for causing the performance differences among the three catalysts. In the alkaline solution, *H species are generated from the water dissociation reaction[22,50]. The KIE tests and in situ Raman measurements (Fig. 4a–c) have demonstrated that Fe NPs play an important role in facilitating water dissociation and feeding the *H species, so Fe-poN-C/Fe exhibits superior performance after introducing Fe NPs. To further confirm the role of Fe NPs on the *H supply, we calculated the free energy diagram for dissociative water reaction and HER on Fe NPs (Fe13 model, Fig. S24a) and graphite carbon (C model, Fig. S24b)[22,49]. It can be seen from Fig. S24c, d that both the energies of H₂O dissociation and *H formation are significantly smaller on Fe13 than those on C, the presence of Fe NPs helps facilitate H₂O dissociation, which is quite possible to enhance the *H coverage on the catalysts surface and then promote

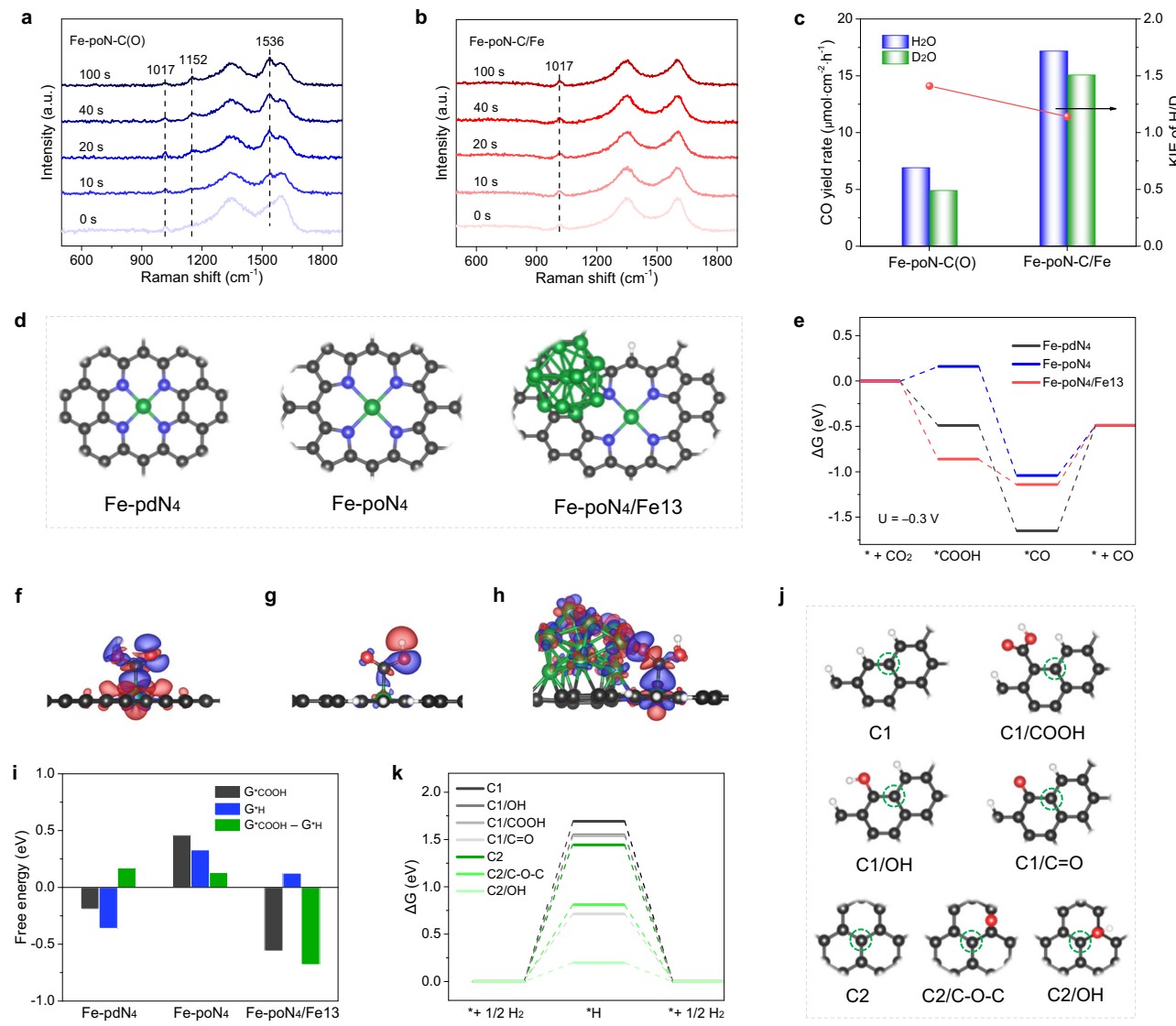

**Fig. 5 | Mechanism investigation using in situ Raman, KIE measurements and DFT calculations.** In situ Raman spectra of (**a**) Fe-poN-C(O) and (**b**) Fe-poN-C/Fe. (**c**) KIE values and CO yield rates of Fe-poN-C(O) and Fe-poN-C/Fe. **d** The optimized structures of Fe-pdN₄, Fe-poN₄ and Fe-poN₄/Fe13. **e** Free energy diagrams of ECR at −0.3 V. Differential charge distribution on (**f**) Fe-pdN₄, (**g**) Fe-poN₄ and (**h**) Fe-poN₄/ Fe13 with adsorption of *COOH. **i** The free energy for the formation of *COOH (G*COOH) or *H (G*H), and the difference between G*COOH and G*H. (**j**) The optimized graphene structure without and with different oxygen-containing groups. **k** Free energy diagrams of HER on graphene structure without and with different oxygenated groups.

$CO_2$ reduction on active sites[41,50]. In addition, Fe is usually active for the side hydrogen evolution reaction (HER), while the Fe NPs in Fe-poN-C/ Fe does not promote side HER, such case may be caused by the size effect and metal-carbon support interaction (Fig. S25)[51,52].

The Gibbs free energy diagram for HER was calculated and presented in Figure S22. The free energy change required for the *H formation of Fe-poN₄ (0.33 eV) is higher than that of Fe-pdN₄ (−0.36 eV) and Fe-poN₄/Fe13 (0.12 eV), indicating the key role of Fe-poN₄ sites in inhibiting HER and improving CO selectivity. Though Fe-poN₄/ Fe13 shows an optimal HER energetics, a much lower value (−0.68 eV) of the difference between the free energy change of G*COOH and G*H than that of Fe-pdN₄ (0.17 eV) and Fe-poN₄ (0.13 eV) was achieved (Fig. 5i), well explained the superior CO selectivity of Fe-poN-C/Fe shown in Fig. 3b[16,47,53]. Furthermore, the activity of different types of oxygenated groups (carboxyl group (−COOH), hydroxyl group (−OH), carbonyl group (−C = O) and epoxy group (−C−O−C)) on the edged (C1) and central (C2) positions of graphene structure in HER was explored through DFT calculations (Fig. 5j). Obviously, the graphene

structure with oxygenated groups shows lower free energy change for *H formation than perfect graphene structure (Fig. 5k), indicating the oxygen-containing groups on carbon surface are unfavorable for ECR process. Here, we believe that it is necessary to reduce the content of oxygen species on the carbon supports of M-N-C catalysts, which can further improve the ECR selectivity.

In summary, a quite energy-efficient ECR catalysts (Fe-poN-C/Fe) was successfully synthesized, which consisting of Fe NPs and pyrrole-type Fe-N₄ sites supported by less-oxygenated carbon matrix according to a series of structural characterizations. Such a hybrid electrocatalyst provides a high FE above 99% for $CO_2$-to-CO under a low overpotential of 0.24 V in an H-type cell. A high CEE over 80% maintain under a potential range of −0.12 to −0.6 V (with *iR* correction) in a flow cell, and a maximum CEE of 97.1% with nearly 100% FE_CO and a current density of −14.1 mA·cm⁻² was obtained at an ultralow overpotential of 21 mV, this value is highest to date. Importantly, Fe-poN-C/Fe exhibits a durable stability test over 100 h accompanying with a high CEE (>90%) and a CO selectivity nearly

100% at a current density over 40 mA·cm$^{-2}$. In situ Raman measurements and the KIE investigations show that the Fe NPs boosts the proton transfer from $CO_2$ to *COOH and sequent reducing the overpotential, the insight is also confirmed by DFT calculations. Meanwhile, it is found that constructing pyrrole-type Fe-N$_4$ sites and limiting oxygen species on carbon supports can well suppress the HER and then improve CO selectivity according to various control experiments and DFT calculations. This work can further nudge the industrial application of ECR and provide a guidance for the development of efficient M-N-C catalysts used for other catalytic reactions.

## Methods

### Chemicals

Iron(III) tetraphenylprophyrin was purchased from Beijing Innochem Science & Technology Co. Ltd., Iron(III) chloride hexahydrate was purchased from Heowns Co. Ltd., and 1,10-phenanthroline was purchased from Ark Pharm Co. Ltd. All reagents were used without additional purification.

### Preparation of catalysts

**Preparation of Fe-poN-C/Fe.** 0.048 mmol of Iron(III) tetraphenylprophyrin (FeTpp) was first dissolved in 3 mL of N,N-Dimethylformamide. Next, 120 mg of carbon black (Vulcan XC-72R, CB) was added into the above solution and sonicated for 0.5 h. The mixture was stirred at 50 °C for 5 h and then evaporated at 80 °C. The resulting powder was heated at 700 °C (5 °C·min$^{-1}$) for 2 h under a mixed hydrogen (5%)/argon atmosphere to obtain Fe-poN-C/Fe.

**Preparation of Fe-pdN-C(O) and Fe-poN-C(O).** First, 0.048 mmol of Iron(III) chloride hexahydrate and 0.144 mmol of 1,10-phenanthroline were mixed in 5 mL of ethanol and stirred at room temperature for 1 h to obtain the Fe-phen complex. Then 120 mg of CB was added to the above complex solution. The mixture was sonicated for 0.5 h, stirred at 50 °C for 5 h, and evaporated at 80 °C. Finally, the resulting powder was heated at 600 °C (5 °C·min$^{-1}$) for 2 h under an argon atmosphere to obtain Fe-pdN-C(O). Fe-poN-C(O) was prepared using the same method as Fe-poN-C/Fe, only with different pyrolysis temperature (600 °C) and atmosphere (argon).

**Preparation of N1-C and N2-C.** The N1-C was obtained by the same procedure as Fe-pdN-C(O) except without the addition of Iron(III) chloride hexahydrate, N2-C was obtained by the same procedure as Fe-poN-C(O) except that FeTpp was replaced by tetraphenylprophyrin.

**Preparation of Fe-poN-C/Fe-(H$_2$SO$_4$/H$_2$O$_2$).** According to the method of the reported literature[23], 100 mg of Fe-poN-C/Fe was dispersed in 150 mL of 0.5 M H$_2$SO$_4$ solution, heated up to 90 °C and stirred for 3 h. Then, 30 mL of 30% H$_2$O$_2$ solution was added slowly to the above solution and stirred for 12 h. After that, the resulting powder was rinsed with water until pH = 7, dried at 80 °C to obtain Fe-poN-C/Fe-(H$_2$SO$_4$/H$_2$O$_2$).

### Characterizations

The morphology of the samples was observed by transmission electron microscope (TEM, FEI Talos F200X G2, AEMC). High-angle annular dark-field scanning transmission electron microscopy (HAADF-STEM) measurements were performed on a JEM-ARM200F with a probe corrector. Powder X-ray diffraction (XRD) patterns were performed on a SmartLab 9KW with Cu Kα radiation. X-ray photoelectron spectroscopy (XPS) analysis was performed on a Thermo Scientific ESCALAB 250Xi at Shiyanjia lab (www.Shiyanjia.com). The contents of metals in prepared samples were measured by inductively coupled plasma-optical emission spectrometer (ICP-OES, Spectro-Blue). The X-ray absorption fine structure spectra (Fe K-edge) were acquired at the 1W1B station in the Beijing Synchrotron Radiation Facility (BSRF). The storage rings of BSRF were operated at 2.5 GeV with an average current of 250 mA.

### Preparation of working electrode

In an H-type cell, the mixture containing 5 mg of catalyst, 20 μL of Nafion solution (5 wt%) and 0.98 mL of ethanol was sonicated for 2 h, and then 100 μL of the above catalyst ink was deposited on carbon paper with a catalyst loading of 0.5 mg·cm$^{-2}$. In a flow cell, 5 mg of the catalyst and 20 μL of Nafion solution (5 wt%) were mixed into 0.98 mL of water-ethanol (1:3 vol) solution, sonicated for 2 h to form a homogeneous ink, and then drop-cast onto a carbon gas-diffusion electrode (GDE, Sigracet 29BC) with a catalyst loading of 1 mg·cm$^{-2}$.

### Electrochemical measurements

In an H-type cell, $CO_2$ electroreduction was carried out on a sealed two-compartment cell separated by a Nafion 117 membrane. An Ag/AgCl electrode (saturated KCl) and Pt plate were utilized as the reference electrode and counter electrode, respectively. All the electric potential potentials in this work were converted to versus reversible hydrogen electrode (vs. RHE) by the equation ($E_{RHE} = E_{Ag/AgCl} + 0.0591 \times$ pH + 0.210 V) unless otherwise specified. Linear sweep voltammetry (LSV) was performed in $CO_2$-saturated 0.5 M KHCO$_3$ aqueous solution with a scan rate of 20 mV·s$^{-1}$. The electrolyte in the cathode chamber was purged with high-purity $CO_2$ at 50 mL·min$^{-1}$ for 30 min before electrolysis. The performance of $CO_2$ electroreduction at different potentials was evaluated by chronoamperometry (CA) in $CO_2$-saturated 0.5 M KHCO$_3$ aqueous solution with rapid stirring. Electrochemical impedance spectroscopy (EIS) measurement was carried out by applying an AC voltage of 5 mV amplitude in a frequency range from 100 kHz to 10 mHz. In a flow cell, 1 M KOH aqueous solution was the electrolyte with nickel mesh, Ag/AgCl and catalyst-coated GDE as the counter electrode, reference electrode, and working electrode, respectively. The anodic and cathodic chambers were separated by a Nafion 117 membrane. The flow rate of $CO_2$ was 28 mL·min$^{-1}$, and the flow rate of the catholyte was 5 mL·min$^{-1}$. The gas products were examined using an online gas chromatograph (GC9790Plus, FULI INSTRUMENTS) and the liquid products were detected using Bruker AVANCE III 400 MHz nuclear magnetic resonance (NMR). The Faradic efficiency of CO (or H$_2$) generation was acquired by the following equation:

$$FE = \frac{z \times F \times P \times a \times L}{j \times R \times T} \qquad (1)$$

where $z$ represents the number of electrons exchanged for gas product formation (z is 2 for CO or H$_2$), $F$ is the Faradaic constant (96,485 C·mol$^{-1}$), $P$ is the atmospheric pressure (1.01 bar), $\alpha$ is the concentration of gas products determined by GC, $L$ is the volume flow rate of the $CO_2$, $j$ is the total current, $R$ is the gas constant (8.314 J·mol$^{-1}$·K$^{-1}$), $T$ is the room temperature.

The turnover frequency (*TOF*, h$^{-1}$) for CO generation was calculated as follows:

$$TOF = \frac{j_{CO}/n \times F}{\omega \times m_{cata}/M} \qquad (2)$$

where $j_{co}$ represents the partial current of CO, $\omega$ is the content of Fe in the catalyst acquired from ICP-OES, $m_{cata}$ represents the catalyst loading (0.5 mg·cm$^{-2}$), and $M$ represents the atomic mass of Fe (55.845 g·mol$^{-1}$).

The cathodic energetic efficiency (*CEE*) for CO generation in a flow cell was calculated as follows:

$$CEE = \frac{E_{cell} \times FE_{CO}}{1.23 - E_{cathode}} \qquad (3)$$

Where $E_{cell}$ represents the thermodynamic cell potential between cathode and anode reactions, which is $E_{cell} = 1.23 - (-0.109) = 1.339$ V, 1.23 V and $-0.109$ V are the thermodynamic potentials for water oxidation and ECR to CO, respectively; $E_{cathode}$ is the applied potential after an $iR$ compensation, $E_{cathode} = E_{RHE} + 0.85 \times i \times Rs(1.68\ \Omega)$.

## In situ Raman measurements

Electrochemical in situ Raman experiments were performed on a custom Raman cell in conjunction with a confocal Raman spectrometer (HORIBA XploRA PLUS). The catalyst ink was loaded onto the glassy carbon to form a catalyst layer. A 532 nm excitation laser (50%) served as excitation source and focused on the catalyst surface through a 50× long working distance objective. All measurements were conducted under ambient conditions and the Raman spectra were recorded at $-0.35$ V in $CO_2$-saturated 0.5 M $KHCO_3$ solution by collecting 5 accumulations at an acquisition time of 2 s.

## DFT calculations

Density functional theory (DFT) as implemented in the Vienna Ab-initio Simulation Package (VASP) was used for all the spin calculations[54,55]. The generalized gradient approximation (GGA) in the form of the Perdew-Burke-Ernzerhof functional (PBE) was adopted to describe the exchange-correlation interactions[56]. A cut-off energy is 450 eV for plain-wave basis sets and the convergence threshold are $10^{-5}$ eV and $3 \times 10^{-5}$ eV/Å for energy and force, respectively. The van der Waals interaction was calculated by the DFT + D3 method using empirical correction in Grimme's scheme[57]. The vacuum slab was set to be more than 15 Å and k point was set to $3 \times 3 \times 1$ during the calculation. All transition barrier calculations were performed using the climbing image nudge elastic band method (CI-NEB)[58].

The Gibbs free energy ($\Delta G$) for each reaction process can be given by:

$$\Delta G = \Delta E_{DFT} + \Delta E_{DFT} + \Delta E_{ZPE} - T\Delta S$$

where $\Delta E_{DFT}$, $\Delta E_{ZPE}$ and $\Delta S$, represent the changes in the DFT total energy, the zero-point energy, and the entropy at 298.15 K, respectively.

The computational hydrogen electrode model (CHE) developed by Nørskov *et al.* was adopted to calculate the free energy of reactions[59,60]. By this model, the applied potential was obtained by the proton-electron pair based on the correlation between chemical and electrical potential, $\Delta G = -eU$, where $e$ is the elementary positive charge, and $U$ is the applied potential.

## Data availability

The main data supporting the findings of this study are available within the article and its Supplementary Information or are available from the corresponding authors upon reasonable request. Source data for Figs. 1–5 and DFT calculation results have been deposited in the Figshare database under accession code (https://doi.org/10.6084/m9.figshare.23713836)[61]. Source data are provided with this paper.

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

## Acknowledgements

This work was financially supported by the National Natural Science Foundation of China with grant number of 22172082 (W.L.), 21978137 (Q.X.G.), and 21878162 (W.L.). This project was also financially funded by the Natural Science Foundation of Tianjin with a grant number of 20JCZDJC00770 (W.L.). The authors would like to thank Fanghui Wang from Shiyanjia Lab (www.shiyanjia.com) for the XPS analysis.

## Author contributions

C.W. developed the conceptual idea and performed the whole experiments. X.Y.W. and H.A.R. worked on material characterizations and electrocatalysis experiments. X.M.Z. and Y.L.Z. carried out DFT calculations. J.W. carried out in situ Raman. Q.X.G. and Y.P.L. commented on the paper. C.W. wrote the paper. W.L. advised and supervised the work.

## Competing interests

The authors declare no competing interests.
