## [Peer Review File · Nature Communications]

Combining Fe nanoparticles and pyrrole-type Fe-N₄ sites on less-oxygenated carbon supports for electrochemical CO₂ reductionREVIEWER COMMENTS

Reviewer #1 (Remarks to the Author):

This manuscript by Wang et al. reports a modified Fe-N-C catalyst with less oxygen content and Fe nanoparticles formed via H₂ reduction treatment. This catalyst shows better CO₂ reduction performance than normal Fe-N-C with pyrrole-type or pyridine-type Fe-N₄ sites. The authors claim that introducing Fe nanoparticles can accelerate the proton transfer from CO₂ to *COOH and lower the free energy for *COOH formation. The findings presented in this work are interesting, but much more efforts should be input to improve its quality for publication.

1. Data reporting.

The authors report a high cathode energy efficiency (CEE) of 97.1%, and 90% CEE during stability test for 40h. It is important to indicate at which current density these values are obtained. In principle, the lower current density, the higher energy efficiency. Furthermore, I do not feel that it makes much sense to use iR-corrected potentials in the calculations of energy efficiency, even for half/cathode energy efficiency. Such concerns also apply to Figure 3c. Comparing only overpotential is not a meaningful way. Considering these metrics, the reported performance enhancement is not that significant.

How are the FEs calculated? It seems that the sum FEs are very close to 100%. Did the authors normalize to 100%?

2. Another concern is the role of Fe nanoparticles. Fe is usually active for the side hydrogen evolution reaction. It is not well explained why it is not the case in this work. DFT calculations are insufficient. Fe nanoparticles might be lost during reaction. It is recommended to quantify the amount of Fe before and after reaction.

Line 245-248: The control experiments for removing Fe nanoparticles and increasing oxygen content should be conducted separately.

Can the oxygen species be reduced under negative potential control?

3. Line 298-300: "while no peaks corresponded to *CO₂⁻ were observed on Fe-poN-C/Fe, implying that the *CO₂⁻ can be speedily protonated to the key *COOH intermediates on Fe-poN-C/Fe." This statement is not appropriate. The absence of an intermediate does not necessarily mean its fast conversion. Thus, the subsequent discussion "Fe NPs in promoting the proton transfer process" is not convincing. What is the applied current/potential in the in situ Raman measurements? Can they see the *CO₂⁻ at a lower current/potential?

Reviewer #2 (Remarks to the Author):

Wang et al. developed a highly selective and energy efficient catalyst for electrochemical CO₂ reduction to CO by testing different families of Fe-N-C-based catalysts. Two different precursors, i.e. Fe(III)-phenanthroline and Fe(III)-tetraphenylporphyrin (FeTpp), underwent pyrolysis for 600 °C in Ar atmosphere to form Fe-pd-NC(O) and Fe-poN-C(O) candidates. Besides, Fe(III)-tetraphenylporphyrin was also treated with pyrolysis at 700 °C in H₂/Ar(5:95) atmosphere to evolve in the Fe-poN-C/Fe sample. Such sample presented the lowest O content (1.72% through XPS), yet similar electrochemically active surface area and Fe content as the previous two. Remarkably, albeit the similar properties as the first two samples, Fe-poN-C/Fe enables a significant improvement of the intrinsic activity toward CO generation, delivering 99.7% faradaic efficiency at 0.24 V overpotential on a H-cell (0.5 KHCO₃ electrolyte) and high energy efficiency (97.1%), selectivity (100%), and long-term stability (100 h) in a flow cell (1 KOH electrolyte). According to the authors, three main properties explain such outstanding performance: (1) the presence of pyrrole-type Fe-N₄ active sites, (2) the reduction of overpotential due to neighboring Fe nanoparticles, (3) the removal of oxygen species from carbon supports.

Although the experimental work is novel and impressive, in my opinion it currently lacks a clear understanding of the phenomenon. The spectroscopic and density functional theory studies do not provide sound explanation of the underlying processes nor suggest how to further improve the system

(or if such approach is extendable to other materials). Before acceptance to Nat. Commun. I think the authors should put additional effort in clearly understanding the reasons behind such remarkable experimental results. Below, some of the points which the authors should tackle.

MAJOR POINTS

(1) The Fe-poN-C/Fe sample enables a lower Tafel slope, thus “indicating the CO production on Fe-poN-C/Fe proceed with a faster kinetics as compared with other two catalysts”. No insight on the reaction kinetic was achieved through density functional theory simulations, which instead focus only on thermodynamics. The authors should assess activation energies for the three catalysts along the CO₂ reduction pathway to look for potential differences.

(2) Nyquist plots indicate that the Fe-poN-C/Fe enabled “higher electronic conductivity and easier charge transfer” than the other two samples. The authors should estimate the ohmic resistance from such plots and compare these to standard values from other M-C-N. Such comparison would help them in assessing the impact of such improved conductivity on very high energy efficiency. Besides, how do the authors explain such difference in electronic conductivity? The atomic content of Fe among the three samples is comparable so this should be due to the increased presence of Fe.

(3) What is the role of Fe NPs? The authors first suggest that they facilitate water dissociation through isotope kinetic effect studies. Does this mean that protons are locally generated on Fe NPs and then act as a reservoir for enabling CO₂ reduction on Fe-N₄? If this is the case, this hypothesis should be further investigated through DFT studies. Besides, the authors put forward that FE NPs enhance the electron transfer to the *COOH by calculating Bader charges through DFT simulations. In my opinion this is just a spurious result due to the essence of the DFT formalism employed. The authors carried out constant-charge DFT. Thus, the presence of additional adsorbates in the simulation cell (such as the Fe₁₃ cluster) determines a shift of the Fe site work function and the consequent increase of the COOH Bader charge (see J. Phys. Chem. Lett. 2016, 7, 1686–1690.

<https://doi.org/10.1021/acs.jpcllett.6b00382>; J. Phys. Chem. Lett. 2015, 6, 2663–2668.

<https://doi.org/10.1021/acs.jpcllett.5b01043>). If the authors include an additional electron in the Fe-pdN₄ and Fe-poN₄ cells by varying the VASP tag NELECT or adding K as an electron donor, I expect similar values of COOH Bader charges as in the Fe-poN₄/Fe₁₃ case.

(4) How was the role of applied electric potential included in the Gibbs free energy diagrams (e.g. Fig.5e)? No mention to the computational hydrogen electrode (CHE) is given in the main text nor the SI, thus it is unclear how the authors account for the change in applied potential. If they uses the CHE, the authors should properly acknowledge previous work by Nørskov and co-workers (see J. Phys. Chem. Lett. 2016, 7, 1686–1690. <https://doi.org/10.1021/acs.jpcllett.6b00382>; Energy Environ. Sci. 2010, 3, 1311–1315. <https://doi.org/10.1039/c0ee00071j>). How were the limiting potentials reported in Fig. 5i calculated? According to Nørskov et al. (Energy Environ. Sci. 2010, 3, 1311–1315. <https://doi.org/10.1039/c0ee00071j>) “The least-negative potential at which the pathway to each product becomes exergonic (downhill in free energy) is referred to as the limiting potential”. I observe > 0.5 eV energy required to desorb *CO from any of the three catalysts (Fig.5e) so I do not understand how these simulations support such low limiting potentials (Fig. 5i) and experimental results of almost full CO selectivity. Such strong CO binding should instead indicate CO poisoning, so very low CO F.E. and conversely high hydrogen evolution rates.

(5) While the authors focus on the CO₂ vibrational bands, I suggest them to look for the 250 cm⁻¹/330 cm⁻¹ and 1800cm⁻¹/2100 cm⁻¹ regions (CO vibrational bands). The relative intensities of these peak provide nice insights on the CO species (see J. Am. Chem. Soc. 2022, DOI:

[10.1021/jacs.2c03172](https://doi.org/10.1021/jacs.2c03172). <https://doi.org/10.1021/jacs.2c03172>; Angew. Chem. Int. Ed. 2021, 60, 16576–16584. <https://doi.org/10.1002/anie.202104114>; ACS Catal. 2021, 11, 7694–7701.

<https://doi.org/10.1021/acscatal.1c01478>.

MINOR POINTS

(1) Please mention that all the electric potential in the main text are given vs RHE. To the best of my knowledge, this is never mentioned.

(2) Few sentences / typos should be corrected, for instance “As a result, there is an urgent demand to excavate and screen an advanced electrocatalyst for energy-efficient ECR, (with the) core challenge (of reducing) overpotential [...]”; “An apparent decline in the FECO and jCO of Fe-poN-C/Fe-

(H₂SO₄/H₂O₂) w(as) observed (Figure. S16), (which) reveals that the Fe NPs possess a positive role"; "To deeply understand (how) the local structure, coordination N type, Fe NPs and the oxygen species on carbon (contributes to) the performance of Fe-N-C catalysts on ECR to CO,"; "The (insight) is also confirmed by DFT calculations.

(3) Remove unformal English words, for instance "What's more,".

Reviewer #3 (Remarks to the Author):

In this work, Wang et al. reported a catalyst consisting of pyrrole type Fe-N₄ sites and Fe nanoparticles supported by less-oxygenated carbon matrix. Based on the experimental and theoretical studies, pyrrole type Fe-N₄ sites improved CO selectivity and Fe nanoparticles boosted the proton transfer. As a result, the catalyst showed a high CO FE above 99% under a low overpotential, and a high CEE over 80% maintain under a potential range of -0.12 to -0.6 V in a flow cell. However, I believe the conclusion cannot be fully supported by the authors' experimental and theoretical results. There are many concerns the authors need to clarify. Overall, I do not feel this manuscript meets the threshold for publication in Nature Communications.

1. The type of coordination N in Fe-pdN-C(O) and Fe-poN-C(O) is speculated through precursor type, without experiment evidences (Line 140-143).
2. The high-resolution N 1s spectra should be fitted by different N species. Additionally, the Fe-N bond length in Fe-poN-C/Fe is shorter than that in Fe-poN-C(O), the authors think that derive from the strong interaction between Fe NPs and Fe-N₄ sites, while the Fe-N binding energy in N 1s spectra of Fe-poN-C/Fe and Fe-poN-C(O) is same, which seems contradictory.
3. All electrochemistry figures lack error bars.
4. In SCN⁻ poison experiments, why SCN⁻ cannot poison Fe nanoparticles (Lines 240-242)? What is the mechanism of SCN⁻ poison?
5. The authors treated Fe-poN-C/Fe catalyst with 0.5 M H₂SO₄ containing certain 30% H₂O₂ to remove Fe NPs and increase O species content simultaneously. It is not sufficient to confirm the role of Fe NPs and O species, due to the presence of two variables (Lines 246-255).
6. The 0.5 M KHCO₃ electrolyte was used in H type cell, why the electrolyte in flow cell was changed to 1 M KOH aqueous (Line 264)?
7. The stability test was performed in H type cell or flow cell (Lines 285-286)? If it was in flow cell, the current density is too low.
8. The figure numbers of Figure 5 were wrong in the manuscript. Line 319, the authors think the RDS on Fe-poN₄ is the *COOH formation, however, it is obvious that the RDS on Fe-poN₄ in Figure 5e is *CO desorption. Please show the free energy change value.

Response to Reviewers

Reviewer #1 (Remarks to the Author):

This manuscript by Wang et al. reports a modified Fe-N-C catalyst with less oxygen content and Fe nanoparticles formed via H₂ reduction treatment. This catalyst shows better CO₂ reduction performance than normal Fe-N-C with pyrrole-type or pyridine-type Fe-N₄ sites. The authors claim that introducing Fe nanoparticles can accelerate the proton transfer from CO₂ to *COOH and lower the free energy for *COOH formation. The findings presented in this work are interesting, but much more efforts should be input to improve its quality for publication.

Response: We are grateful to the reviewer for such kind and valuable comments.

Comment 1: Data reporting. The authors report a high cathode energy efficiency (CEE) of 97.1%, and 90% CEE during stability test for 40h. It is important to indicate at which current density these values are obtained. In principle, the lower current density, the higher energy efficiency. Furthermore, I do not feel that it makes much sense to use iR-corrected potentials in the calculations of energy efficiency, even for half/cathode energy efficiency. Such concerns also apply to Figure 3c. Comparing only overpotential is not a meaningful way. Considering these metrics, the reported performance enhancement is not that significant.

How are the FEs calculated? It seems that the sum FEs are very close to 100%. Did the authors normalize to 100%?

Response: Thank the reviewer for the pertinent comments and questions. In our study, Fe-poN-C/Fe exhibits a high CEE of 97.1% with a current density of $-14.06 \text{ mA}\cdot\text{cm}^{-2}$, which

also can maintain a > 90% CEE with a current density over 40 mA·cm⁻² during stability test for 100 h (*please see line 303-305 in the manuscript*). The goal of this work is to achieve highly energy-efficient ECR by reducing overpotential and improving selectivity. We understand the reviewer's concerns about the current density. Unlike increasing the current density of the catalysts, reducing overpotential and improving selectivity are full of challenges, as they are intrinsic properties of the catalysts. The current density can be indeed improved by the optimization process such as the cell design, the catalyst loading, the choice of electrolyte, and the choice of catalyst support. Furthermore, we agree that it is not significant to use *iR*-corrected potentials in the calculations of cathode energy efficiency. Therefore, the FE_{CO} of Fe-poN-C/Fe at different overpotentials with and without *iR* correction were all given in Figure 4c. It can be seen that Fe-poN-C/Fe possesses an outstanding FE_{CO} over 99% at a low overpotential (Figure 4c), regardless of whether there is *iR* correction or not, this performance outperforms almost all state-of-the-art electrocatalysts for ECR to CO. However, almost all literatures reporting cathode energy efficiency use *iR* correction, so we present the cathode energy efficiency (Figure 4d and Figure S19a) under *iR* correction for fair comparison with the literatures. In addition, the novelty of this work lies in achieving highly energy-efficient ECR (low overpotential and high FE_{CO}) through precisely design and modify Fe-N-C electrocatalysts. Thus, Figure 3c shows the FE_{CO} of Fe-poN-C/Fe at low overpotential and the comparison with other previously reported CO₂-to-CO electrocatalysts.

The Faradic efficiency of CO (or H₂) generation was acquired by the following equation:

$$FE = \frac{n \times z \times F}{Q} = \frac{z \times F}{j \times t} \times \frac{P \times V}{R \times T} = \frac{z \times F}{j \times t} \times \frac{P \times \alpha \times L \times t}{R \times T} = \frac{z \times F \times P \times \alpha \times L}{j \times R \times T}$$

where *n* denotes the moles of the generated CO or H₂, *z* represents the number of electrons

exchanged for gas product formation (z is 2 for CO or H₂), F is the Faradaic constant (96485 C·mol⁻¹), Q is the total charge, j is the total current, t is the electrolysis time, P is the atmospheric pressure (1.01 bar), V is the volume of gas product, R is the gas constant (8.314 J·mol⁻¹·K⁻¹), T is the room temperature, α is the concentration of gas products determined by GC, L is the volume flow rate of the CO₂. In order to better demonstrate the calculation method of Faraday efficiency, we have modified the original equation as follows: “

$$FE = \frac{z \times F \times P \times \alpha \times L}{j \times R \times T}$$

where z represents the number of electrons exchanged for gas product formation (z is 2 for CO or H₂), F is the Faradaic constant (96485 C·mol⁻¹), P is the atmospheric pressure (1.01 bar), α is the concentration of gas products determined by GC, L is the volume flow rate of the CO₂, j is the total current, R is the gas constant (8.314 J·mol⁻¹·K⁻¹), T is the room temperature.” (please see Revised Supplementary Information, Page 6)

Furthermore, we did not normalize Faraday efficiency. As shown in Figure 3b, Figure S7 and Figure R1, the sum FEs are not very close to 100%. In this study, we performed at least three measurements and take the average in order to obtain reliable data. Error bars have been added into all electrochemistry figures containing the Faradic efficiency and current density.

Figure R1. FE of CO and H₂ for (a) Fe-pdN-C(O), (b) Fe-poN-C(O) and (c) Fe-poN-C/Fe.

Comment 2: Another concern is the role of Fe nanoparticles. Fe is usually active for the side hydrogen evolution reaction. It is not well explained why it is not the case in this work. DFT calculations are insufficient. Fe nanoparticles might be lost during reaction. It is recommended to quantify the amount of Fe before and after reaction.

Line 245-248: The control experiments for removing Fe nanoparticles and increasing oxygen content should be conducted separately.

Can the oxygen species be reduced under negative potential control?

Response: Thank the reviewer for the warm questions and comments. The comment can be divided into four interesting questions/concerns, which are discussed separately below.

(1) Fe is usually active for the side hydrogen evolution reaction. It is not well explained why it is not the case in this work.

Indeed, Fe nanoparticles and other metal nanoparticles (Ni, Co and Cu *et al.*) are usually

active for the side hydrogen evolution reaction (HER). However, some recent studies have demonstrated that introducing metal nanoparticles on M-N-C materials not only does not promote HER but also improve ECR selectivity [*Angew. Chem. Int. Ed.* 2021, 60, 11959-119; *Angew. Chem. Int. Ed.* 2021, 60, 24022-24027; *ACS Catal.* 2022, 12, 7517-7523]. Likewise, in our study, the catalytic selectivity of Fe-N-C toward CO production improves especially at low potentials after the introduction of Fe NPs. The size effect of NPs and the metal-carbon support interaction may be the reason for such case [*Angew. Chem. Int. Ed.* 2020, 59, 18572-185; *ACS Catal.* 2022, 12, 7517-7523]. According to the study of Li *et al.* [*Angew. Chem. Int. Ed.* 2020, 59, 18572-18577], three models were built to explore the size of NPs and carbon support on the HER performance of Fe NPs using DFT calculations: 1) Fe (110) represents Fe NPs with large size; 2) Fe₁₃ clusters represents Fe NPs with small size; 3) Fe₁₃C represents Fe NPs after the incorporation of carbon. As shown in Figure R2, the size of Fe NPs has a significant impact on the free energy change for *H formation, and the HER on Fe NPs became difficult after the incorporation of carbon. To clarify this point in our work, we have added Figure R2 as Figure S25 and some new descriptions in the Revised Manuscript: “In addition, Fe is usually active for the side hydrogen evolution reaction (HER), while the Fe NPs in Fe-poN-C/Fe does not promote side HER, such case may be caused by the size effect and metal-carbon support interaction (Figure S25).^{51,52}” (please see line 384-387 in the Revised Manuscript) “Fe (110) represents Fe NPs with large size, Fe₁₃ clusters represents Fe NPs with small size, Fe₁₃C represents Fe NPs after the incorporation of carbon. Obviously, the size of Fe NPs has a significant impact on the free energy change for *H formation, and the HER on Fe NPs became difficult after the incorporation of carbon.” (please see Revised

Figure R2. The model of (a) Fe (110), (b) Fe13 and (c) Fe13C. (d) Free energy diagrams of HER on the models of Fe (110), Fe13 and Fe13C.

(2) Fe nanoparticles might be lost during reaction. It is recommended to quantify the amount of Fe before and after reaction.

We initially plan to quantify the amount of Fe before and after reaction using ICP-OES and further explore whether Fe nanoparticles were lost during reaction. However, this method is impractical because the mass of the catalyst scraped off the carbon paper after reaction is inaccurate (it mixes part of the carbon paper substrate and nafion components). Thus, the following measures were adopted to reply the reviewer's concerns. First, the Fe contents in Fe-poN-C/Fe before and after H_2SO_4 treatment were 1.14 wt% and 0.73 wt%, respectively. This indicates that the contents of Fe NPs in Fe-poN-C/Fe were approximately 0.41 wt%. Secondly, the mass of Fe in the electrolyte was 0.165 μg ($0.011 \mu g \cdot mL^{-1} \times 15 mL$, $0.011 \mu g \cdot mL^{-1}$ was obtained from ICP-OES measurements, 15 mL represents the volume of cathode

electrolyte) after a stability test for 10 h at -0.35 V in a H cell. Given that the mass of total Fe and Fe NPs in the cathode catalyst was $5.7 \mu\text{g}$ ($1.14 \text{ wt}\% \times 0.5 \text{ mg}$, 0.5 mg represents the catalysts loading) and $2.05 \mu\text{g}$ ($0.41 \text{ wt}\% \times 0.5 \text{ mg}$), respectively, so the loss of Fe ($0.165 \mu\text{g}$) is minimal, and Fe nanoparticles are almost not lost during reaction. This is also one of the key factors why Fe-poN-C/Fe can still maintain excellent performance after a stability test over 100 h in a flow cell (Figure 4e). Additionally, Figure R3 (as Figure S20 in the supporting information) show that Fe NPs are well preserved after the long-term testing.

Figure R3. (a) HAADF-TEM, (b,c) aberration-corrected HAADF-STEM, and (d) EDS mapping of Fe-poN-C/Fe after long-term electrolysis.

(3) The control experiments for removing Fe nanoparticles and increasing oxygen content should be conducted separately.

As suggested by the reviewer, the control experiments for removing Fe nanoparticles and increasing oxygen content have been conducted separately. In order to confirm the effect of

oxygen species on the ECR performance, Figure R4 present the full XPS spectra, FE_{CO} and j_{CO} of Fe-poN-C(O) before and after H_2O_2/H_2SO_4 treatment. Likewise, Figure R5 present the TEM spectra, FE_{CO} and j_{CO} of Fe-poN-C/Fe before and after H_2SO_4 treatment to investigate the effect of Fe NPs. The results and analysis of the new experiments have added to the Revised Manuscript: “The effect of Fe NPs and oxygen species on the ECR performance was further studied by control experiments. Firstly, Fe-poN-C(O) catalyst was treated with H_2O_2/H_2SO_4 solutions (0.5 M H_2SO_4 containing certain 30% H_2O_2) under 80 °C for 24 h to increase the content of oxygen species. XPS results exhibit that more oxygen species have been introduced on Fe-poN-C(O) after H_2O_2/H_2SO_4 treatment (Figure S15a). The FE_{CO} and j_{CO} of the treated Fe-poN-C(O) (named as Fe-poN-C(O)-(H_2O_2/H_2SO_4)) decreased significantly as compared with Fe-poN-C(O) (Figure S15b,c), indicating the oxygen species on carbon supports are unfavorable for promoting ECR. Secondly, Fe-poN-C/Fe catalyst was treated with H_2SO_4 solutions (0.5 M H_2SO_4) under 80 °C for 24 h to remove the Fe NPs. The HAADF-STEM images of the treated Fe-poN-C/Fe (named as Fe-poN-C/Fe-(H_2SO_4)) show that the Fe NPs are almost all removed and single-atom Fe sites remained (Figure S16a,b). Performance tests exhibits the FE_{CO} at low potentials and j_{CO} decreased after the removal of Fe NPs (Figure S16c,d), which indicates that the existence of Fe NPs is benefit for reducing overpotential on ECR. Furthermore, the HAADF-STEM images and XPS results shown in Figure S17a-c reveal the Fe NPs in Fe-poN-C/Fe are removed while more oxygen species are introduced after H_2O_2/H_2SO_4 treatment. An apparent decline in the FE_{CO} and j_{CO} of Fe-poN-C/Fe-(H_2SO_4/H_2O_2) was observed (Figure. S17d,e), combined with the above two control experiments, which means that the Fe NPs possess a positive role in reducing overpotential

and oxygen species exhibits a negative role in improving CO selectivity.” (please see line

253-273 in the Revised Manuscript)

Figure R4. (a) Full XPS spectra, (b) FE_{CO} and (c) j_{CO} of Fe-poN-C(O) before and after $\text{H}_2\text{O}_2/\text{H}_2\text{SO}_4$ treatment.

Figure R5. (a) HAADF-TEM and (b) aberration-corrected HAADF-STEM of Fe-poN-C/Fe after H₂SO₄ treatment. (c) FE_{CO} and (d) j_{CO} of Fe-poN-C/Fe after H₂SO₄ treatment.

(4) Can the oxygen species be reduced under negative potential control?

For the last question, we originally intend to confirm that whether the oxygen species were reduced under negative potential control by testing the XPS spectra of catalysts before and after ECR, but this method is full of difficulties and impractical due to the presence of nafion components (contained a large amount of oxygen elements) in the tested catalysts. In fact, the reduction of the oxygen species under high negative potential control are possible, but far from probable at low negative potential range. Different oxygen-containing functional groups have different reduction potential. It is reported that the epoxide group (–C–O–C) is considered as the most easily reduced oxygen-containing functional group, while the reduction potential of aromatic –C–O–C still up to –0.75 to –1.5 V vs. SCE in aqueous solutions [*Chem. Eng. J.* 2014, 251, 422-434; *Small* 2011, 7, 1203; *Phys. Chem. Chem. Phys.*

2011, 13, 9187]. Considering that our work mainly focuses on evaluating and exploring the performance differences of studied catalysts within the low potential range, so the reduction of the oxygen species on studied catalysts is unlikely. Moreover, some reported literatures have studied the carbon materials with oxygen-containing species as cathode catalysts in electrocatalytic reduction reaction [*Adv. Mater.* 2023, 35, 2210658; *Nat. Commun.* 2021, 12, 5265; *J. Am. Chem. Soc.* 2019, 141, 20451-20459; *ChemElectroChem* 2020, 7, 1-9], which demonstrates the role of oxygen species and the oxygen species will not be reduced during the reaction.

Comment 3: Line 298-300: “while no peaks corresponded to $*CO_2^-$ were observed on Fe-poN-C/Fe, implying that the $*CO_2^-$ can be speedily protonated to the key $*COOH$ intermediates on Fe-poN-C/Fe.” This statement is not appropriate. The absence of an intermediate does not necessarily mean its fast conversion. Thus, the subsequent discussion “Fe NPs in promoting the proton transfer process” is not convincing. What is the applied current/potential in the in situ Raman measurements? Can they see the $*CO_2^-$ at a lower current/potential?

Response: We thank the reviewer for these warm comments. In the work of Liu *et al.* [*Chem* 2021, 7, 1-11], the authors performed *in situ* ATR-SEIRAS measurements and found that the $*CO_3H_2$ ($*CO_2^- \cdots H_2O$) was not detected on the studied catalysts (FeN₄ sites with surrounding graphitic N), they thought these results can demonstrate that $*CO_2^-$ are rapidly protonated to form $*COOH$ due to the activation of H₂O by graphitic N. Therefore, we also provided such statement based on the similar *in situ* test results. However, as the reviewer pointed out, the

statement of "...implying that the $*\text{CO}_2^-$ can be speedily protonated to the key $*\text{COOH}$..." is not appropriate, especially the statement of "can be". We realize that the absence of an intermediate does not necessarily mean its fast conversion, and the conclusion of " $*\text{CO}_2^-$ were speedily protonated to the key $*\text{COOH}$ intermediates" should be deductive rather than deterministic. Thus, the *in situ* Raman results (The assignment of $*\text{CO}_2^-$ peaks on Fe-poN-C(O) appeared after applying potential, while no peaks corresponded to $*\text{CO}_2^-$ were observed on Fe-poN-C/Fe) indicates that the $*\text{CO}_2^-$ are likely to be rapidly protonated to the $*\text{COOH}$ intermediates on the surface of Fe-poN-C/Fe. It is well known that the activation of H_2O plays a crucial role in the protonation process during ECR to CO [*Angew. Chem. Int. Ed.* 2021, 60, 11959-119]. To further explore the effect of Fe Nps in H_2O activation progress during ECR, the investigation on the kinetic isotope effect (KIE) over Fe-poN-C(O) and Fe-poN-C/Fe catalysts were conducted in our work. The results of KIE measurements show Fe-poN-C/Fe exhibits a much lower KIE value (1.14) than that of Fe-poN-C(O) (1.41), which indicates the H_2O activation on Fe-poN-C/Fe catalysts is easy [*Angew. Chem. Int. Ed.* 2021, 133, 12066-12072; *Nat. Commun.* 2019, 10, 1-10]. In other words, the introduction of Fe NPs in Fe-poN-C/Fe promotes the H_2O dissociation, and this is beneficial for accelerating proton transfer process and forming $*\text{COOH}$ intermediates. According to the reviewer's comments, we have corrected the statements about *in situ* Raman and KIE measurements in the Revised Manuscript: "The assignment of $*\text{CO}_2^-$ peaks on Fe-poN-C(O) appeared after applying potential and gradually strengthened with increasing electrolysis time, while no peaks corresponded to $*\text{CO}_2^-$ were observed on Fe-poN-C/Fe, implying that the $*\text{CO}_2^-$ are likely to be rapidly protonated to the $*\text{COOH}$ intermediates on the surface of Fe-poN-C/Fe.¹⁴ It is well

known that the activation of H₂O plays a crucial role in the protonation process during ECR to CO.²¹ Thus, the investigation on the kinetic isotope effect (KIE) over Fe-poN-C(O) and Fe-poN-C/Fe catalysts were conducted to further explore the effect of Fe NPs in H₂O activation progress during ECR.^{21,41,42} As shown in Figure. 5c, the calculated KIE value of Fe-poN-C(O) is 1.41, while Fe-poN-C/Fe shows a much lower KIE value (1.14), indicating that the H₂O activation on Fe-poN-C/Fe catalysts is easy.^{21,41} In other words, combined with the results of *in situ* Raman results, the introduction of Fe NPs in Fe-poN-C/Fe promotes the H₂O dissociation, and this is beneficial for accelerating proton transfer process and forming *COOH intermediates.” (please see line 315-327 in the Revised Manuscript)

The *in situ* Raman spectra of Fe-poN-C(O) and Fe-poN-C/Fe catalysts were recording in CO₂-saturated 0.5 M KHCO₃ under an applied potential of -0.35 V (please see line 310-311 in the manuscript). We measured the Raman spectra of Fe-poN-C(O) catalysts in CO₂-saturated 0.5 M KHCO₃ at different potentials from 0 to -0.5 V with a step interval of 0.1 V. As shown in Figure R6, the *CO₂⁻ peaks can be observed at -0.2 V, which became obvious at -0.3 V and maintained until -0.5 V. Thus, an applied potential of -0.35 V was selected to carry out the *in situ* Raman tests in our work.

Figure R6. *In situ* Raman spectra of Fe-poN-C(O) in CO₂-saturated 0.5 M KHCO₃ at different potentials from 0 to -0.5 V.

Reviewer #2 (Remarks to the Author):

Wang et al. developed a highly selective and energy efficient catalyst for electrochemical CO₂ reduction to CO by testing different families of Fe-N-C-based catalysts. Two different precursors, i.e. Fe(III)-phenanthroline and Fe(III)-tetraphenylporphyrin (FeTpp), underwent pyrolysis for 600 °C in Ar atmosphere to form Fe-pd-NC(O) and Fe-poN-C(O) candidates. Besides, Fe(III)-tetraphenylporphyrin was also treated with pyrolysis at 700 °C in H₂/Ar(5:95) atmosphere to evolve in the Fe-poN-C/Fe sample. Such sample presented the lowest O content (1.72% through XPS), yet similar electrochemically active surface area and Fe content as the previous two. Remarkably, albeit the similar properties as the first two samples, Fe-poN-C/Fe enables a significant improvement of the intrinsic activity toward CO generation, delivering 99.7% faradaic efficiency at 0.24 V overpotential on a H-cell (0.5 KHCO₃ electrolyte) and high energy efficiency (97.1%), selectivity (100%), and long-term stability (100 h) in a flow cell (1 KOH electrolyte). According to the authors, three main properties explain such outstanding performance: (1) the presence of pyrrole-type Fe-N₄ active sites, (2) the reduction of overpotential due to neighboring Fe nanoparticles, (3) the removal of oxygen species from carbon supports.

Although the experimental work is novel and impressive, in my opinion it currently lacks a clear understanding of the phenomenon. The spectroscopic and density functional theory studies do not provide sound explanation of the underlying processes nor suggest how to further improve the system (or if such approach is extendable to other materials). Before acceptance to Nat. Commun. I think the authors should put additional effort in clearly understanding the reasons behind such remarkable experimental results. Below, some of the

points which the authors should tackle.

Response: We are grateful to the reviewer for such kind and valuable comments.

MAJOR POINTS

Comment 1: The Fe-poN-C/Fe sample enables a lower Tafel slope, thus “indicating the CO production on Fe-poN-C/Fe proceed with a faster kinetics as compared with other two catalysts”. No insight on the reaction kinetic was achieved through density functional theory simulations, which instead focus only on thermodynamics. The authors should assess activation energies for the three catalysts along the CO₂ reduction pathway to look for potential differences.

Response: Thank the reviewer for this constructive comment. According to the reviewer’s comment and the literatures [*J. Am. Chem. Soc.* 2018, 140, 11, 3833-3837; *Angew. Chem. Int. Ed.* 2021, 60, 4192-4198], we assessed the activation energies for the three catalysts along the CO₂ reduction pathway through a direct hydrogenation mechanism. As shown in Figure R7, the elementary reactions $*CO_2 + *H + e^- \rightarrow *COOH$ and $*COOH + *H + e^- \rightarrow *CO + *H_2O$ are both exothermic. The activation energy barriers from the $*CO_2$ transition to $*COOH$ on Fe-pdN₄, Fe-poN₄ and Fe-poN₄/Fe13 are 0.19 eV, 0.15 eV and 0.33 eV, respectively, and the activation energy barriers from the $*COOH$ transition to $*CO$ on Fe-pdN₄, Fe-poN₄ and Fe-poN₄/Fe13 are 0.63 eV, 0.24 eV and 0.58 eV, respectively. The activation energies of three catalysts model to form $*COOH$ or $*CO$ are relatively low because the activation energies of typical catalysts (Ni-N-C and Ag) used in ECR to form $*COOH$ or $*CO$ are much higher than 1eV [*J. Am. Chem. Soc.* 2018, 140, 11, 3833-3837;

Angew. Chem. Int. Ed. 2021, 60, 4192-4198]. Therefore, the hydrogenation steps of three catalysts in kinetics during CO₂ reduction are easy on the premise of providing sufficient *H, and the capacity of *H supply before CO₂ hydrogenation step is a key factor for causing the performance differences among the three catalysts. To well reply the reviewer's comments, we have added Figure R7 as Figure S23 and the above analysis in the Revised Manuscript: "To explore the difficulty degree of ECR reaction more actually, the activation energies for the three catalysts model along the CO₂ reduction pathway through a direct hydrogenation mechanism were assessed.^{48,49} As shown in Figure S23, the elementary reactions for *COOH formation and *CO formation are both exothermic, and the activation energies of the three catalysts model to form *COOH (0.15-0.33 eV) or *CO (0.24-0.63 eV) are relatively low because the activation energies of typical catalysts (Ag⁴⁸ and Ni-N-C⁴⁹) used in ECR to form *COOH or *CO are much higher than 1eV. This indicates that the hydrogenation steps of three catalysts during CO₂ reduction are easy on the premise of providing sufficient *H. In other words, the capacity of *H supply before CO₂ hydrogenation step is a key factor for causing the performance differences among the three catalysts." (please see line 364-374 in the Revised Manuscript) "All barriers calculations were performed climbing image nudged elastic band method (CI-NEB)."⁶⁷ (please see Revised Supplementary Information, Page 7)

Figure R7. The activation energies for Fe-pdN₄, Fe-poN₄ and Fe-poN₄/Fe13 along the CO₂ reduction pathway. Insets are simple structures of initial state (IS), transition state (TS) and final state (FS).

Comment 2: Nyquist plots indicate that the Fe-poN-C/Fe enabled “higher electronic conductivity and easier charge transfer” than the other two samples. The authors should estimate the ohmic resistance from such plots and compare these to standard values from other M-C-N. Such comparison would help them in assessing the impact of such improved conductivity on very high energy efficiency. Besides, how do the authors explain such difference in electronic conductivity? The atomic content of Fe among the three samples is comparable so this should be due to the increased presence of Fe.

Response: Thanks for the careful review and questions on the EIS analysis. We have fitted the EIS data with the corresponding equivalent circuit (Figure R8), and Table R1 shows that Fe-poN-C/Fe possesses a lower charge transfer resistance (R_{ct}) than Fe-pdN-C(O) and Fe-poN-C(O), indicating the higher electronic conductivity of Fe-poN-C/Fe and the easier charge transfer during ECR process. Therefore, we have added the equivalent circuit and the value of R_{ct} in Figure S11 as well as corrected the statement about EIS analysis in the Revised Manuscript: “The Nyquist plots show that Fe-poN-C/Fe possesses a lower charge transfer resistance as compared with Fe-pdN-C(O) and Fe-poN-C(O) (Figure. S11), indicating the significantly fast charge-transfer process and improved electronic conductivity after introducing Fe NPs, eventually resulting in an enhanced activity on ECR.” (please see line 238-242 in the Revised Manuscript)

Figure R8. Equivalent circuit for the EIS fitting.

Table R1. Solution resistances (R_s) and charge transfer resistances (R_{ct}) of three catalysts determined by the EIS fitting.

Catalysts	R_s (Ω)	R_{ct} (Ω)
Fe-pdN-C(O)	2.781	425.6
Fe-poN-C(O)	2.724	192.8
Fe-poN-C/Fe	2.732	120.4

Furthermore, we are very sorry that we cannot provide the comparison of R_{ct} of Fe-poN-C/Fe with standard values from other M-N-C materials, because the experimental conditions (such as the type/concentration of electrolyte and the applied potential) for estimating the R_{ct} of M-N-C materials in various literatures are different [Adv. Mater. 2018, 30, 1706617; Nat. Commun. 2019, 10, 2980; Energy Environ. Sci., 2021,14, 2349-2356].

To answer the last question, “how do the authors explain such difference in electronic conductivity?”, the following two explanations are supplied: 1) It is widely known that the conductivity of carbon materials is lower than that of metal materials [Nat. Energy 2021, 6, 1154-1163], so Fe-poN-C/Fe with Fe NPs and Fe-N₄ species exhibits a higher conductivity than that of Fe-pdN-C(O) and Fe-poN-C(O) with only Fe-N₄ species; 2) The introduction of Fe NPs on Fe-poN-C/Fe obviously improved the graphitization degree of the surrounding carbon layer (which can be seen from Figure R9), these highly-graphitized graphene layers

are beneficial for electron transfer, leading to higher conductivity [*Nat. Commun.* 2020, 11, 593].

Figure R9. Aberration-corrected HAADF-STEM of Fe-poN-C/Fe at different positions. It is obvious that the carbon substrate nearby Fe NPs has highly-graphitized graphene layer.

Comment 3: What is the role of Fe NPs? The authors first suggest that they facilitate water dissociation through isotope kinetic effect studies. Does this mean that protons are locally generated on Fe NPs and then act as a reservoir for enabling CO₂ reduction on Fe-N₄? If this is the case, this hypothesis should be further investigated through DFT studies. Besides, the authors put forward that FE NPs enhance the electron transfer to the *COOH by calculating Bader charges through DFT simulations. In my opinion this is just a spurious result due to the essence of the DFT formalism employed. The authors carried out constant-charge DFT. Thus, the presence of additional adsorbates in the simulation cell (such as the Fe₁₃ cluster) determines a shift of the Fe site work function and the consequent increase of the COOH Bader charge (see *J. Phys. Chem. Lett.* 2016, 7, 1686–1690).

<https://doi.org/10.1021/acs.jpcclett.6b00382>; J. Phys. Chem. Lett. 2015, 6, 2663–2668.

<https://doi.org/10.1021/acs.jpcclett.5b01043>). If the authors include an additional electron in the Fe-pdN₄ and Fe-poN₄ cells by varying the VASP tag NELECT or adding K as an electron donor, I expect similar values of COOH Bader charges as in the Fe-poN₄/Fe13 case.

Response: Thank the reviewer for the constructive comments. Based on the results of KIE tests and *in situ* Raman experiments, our study believed that the role of Fe NPs is to accelerate the proton transfer by facilitating the H₂O dissociation, this is beneficial for the formation of *COOH intermediates and thus improve the ECR performance. As the reviewer said, Fe NPs facilitate water dissociation, the protons are likely to locally generate on Fe NPs and then act as a reservoir for enabling ECR on Fe-N₄ sites. To demonstrate this hypothesis, we further performed DFT calculation. We hypothesized that the adjacent sites of Fe-N₄ supply the *H for ECR. Considering that the adjacent sites of Fe-N₄ are Fe NPs or graphite carbon structure, we calculated the free energy diagram for dissociative water reaction and HER on Fe NPs (Fe13 model, Figure R10a) and graphite carbon (C model, Figure R10b) [*Angew. Chem. Int. Ed.* 2021,60, 24022-24027; *Angew. Chem. Int. Ed.* 2021, 60, 4192-4198]. As shown in Figure R10c,d, both the energies of H₂O dissociation and *H formation are significantly smaller on Fe13 than those on C, the presence of Fe Nps helps facilitate H₂O dissociation, which is quite possible to enhance the *H coverage on the catalysts surface and then promote CO₂ reduction on active sites [*Nat. Commun.* 2019, 10, 892; *PNAS* 2022, 119, e2207326119]. We have added Figure R10 as Figure S24 and some new descriptions in the Revised Manuscript: “In the alkaline solution, *H species are generated from the water dissociation reaction.^{22,50} The KIE tests and *in situ* Raman measurements (Figure 4a-c) have

demonstrated that Fe NPs play an important role in facilitating water dissociation and feeding the *H species, so Fe-poN-C/Fe exhibits superior performance after introducing Fe NPs. To further confirm the role of Fe NPs on the *H supply, we calculated the free energy diagram for dissociative water reaction and HER on Fe NPs (Fe13 model, Figure S24a) and graphite carbon (C model, Figure S24b).^{22,49} It can be seen from Figure S24c,d that both the energies of H_2O dissociation and *H formation are significantly smaller on Fe13 than those on C, the presence of Fe NPs helps facilitate H_2O dissociation, which is quite possible to enhance the *H coverage on the catalysts surface and then promote CO_2 reduction on active sites.^{41,50}

(please see line 374-384 in the Revised Manuscript)

Figure R10. The models of (a) C and (b) Fe13. Free energy diagram for (c) dissociative water reaction and (d) HER on the models of C and Fe13.

In recent years, the computational hydrogen electrode (CHE) model has provided an elegant way to explore reaction energetics without explicit treatment of the electrons and ions in solution [*J. Phys. Chem. B* 2004, 108, 17886-17892; *Energy Environ. Sci.* 2010, 3, 1311-

1315; *Angew. Chem., Int. Ed.* 2013, 52, 7282-7285]. Thus, our study and a survey of previous literature (Table R2) calculated the Bader charges and gave such statement (...enhance the electron transfer...) based on the CHE model. However, after carefully reading the references [*J. Phys. Chem. Lett.* 2016, 7, 1686-1690; *J. Phys. Chem. Lett.* 2015, 6, 2663-2668] provided by the reviewer, we know that a major challenge in the ab initio calculation of electrochemical barriers is that simulations are done at constant charge, while real electrochemical systems operate at constant potential, and any consideration of kinetics and charge transfer barriers necessitates the inclusion of the solvent and charge in the model system. We are fully aware of the imperfections of the current Bader charge analysis using CHE model, so the descriptions about Bader charge analysis were deleted in the Revised Manuscript for avoiding the concerns from reviewers and future readers: “Furthermore, three catalyst models binding with the important *COOH intermediate were further investigated by differential charge distribution. As shown in Figure. 5f-h, Fe-poN₄/Fe13 shows an obvious electron density accumulation between the active site and *COOH as compared with Fe-pdN₄ and Fe-poN₄, suggesting that the introduction of Fe NPs is favorable to stabilize the *COOH intermediate, thus boosting the CO₂ reduction process.⁴⁷” (please see line 344-346, 357-359 in the Revised Manuscript)

Table R2. Differential charge distribution with Bader analysis of M-N-C catalysts on other reported literatures.

No.	Differential charge distribution with Bader analysis	Ref.

1		J. Am. Chem. Soc. 2022, 144, 32, 14505-14516
2		Angew. Chem. Int. Ed. 2022, 61, e202206233
3		Nat. Commun. 2022, 13, 57

Comment 4: How was the role of applied electric potential included in the Gibbs free energy diagrams (e.g. Fig.5e)? No mention to the computational hydrogen electrode (CHE) is given in the main text nor the SI, thus it is unclear how the authors account for the change in applied potential. If they uses the CHE, the authors should properly acknowledge previous work by Nørskov and co-workers (see *J. Phys. Chem. Lett.* 2016, 7, 1686–1690. <https://doi.org/10.1021/acs.jpcclett.6b00382>; *Energy Environ. Sci.* 2010, 3, 1311–1315. <https://doi.org/10.1039/c0ee00071j>). How were the limiting potentials reported in Fig. 5i calculated? According to Nørskov et al. (*Energy Environ. Sci.* 2010, 3, 1311–1315. <https://doi.org/10.1039/c0ee00071j>) “The least-negative potential at which the pathway to each product becomes exergonic (downhill in free energy) is referred to as the limiting potential”. I observe > 0.5 eV energy required to desorb *CO from any of the three catalysts (Fig.5e) so I do not understand how these simulations support such low limiting potentials (Fig. 5i) and experimental results of almost full CO selectivity. Such strong CO binding

should instead indicate CO poisoning, so very low CO F.E. and conversely high hydrogen evolution rates.

Response: Thank the reviewer for the warm questions and comments. Considering that there is a significant performance difference among the three catalysts under the low applied potential range ($-0.25\text{ V} \sim -0.45\text{ V}$) in Figure 3, we show the Gibbs free energy diagrams at an applied electric potential (-0.3 V) in Figure 5e to better combine with the experimental section. We indeed obtained the free energy change under an applied electric potential of -0.3 V using the computational hydrogen electrode (CHE) model. Therefore, we have added some new descriptions in the DFT calculations section and cited the classic literatures of Nørskov and co-workers [*Energy Environ. Sci.* 2010, 3, 1311-1315; *J. Phys. Chem. Lett.* 2016, 7, 1686-1690]: “The computational hydrogen electrode model (CHE) developed by Nørskov *et al.* was adopted to calculate the free energy of reactions.^{7,8} By this model, the applied potential was obtained by the proton-electron pair based on the correlation between chemical and electrical potential, $\Delta G = -eU$, where e is the elementary positive charge, and U is the applied potential.” (please see Revised Supplementary Information, Page 7)

As the reviewer stated, the studies of Nørskov *et al.* [*Energy Environ. Sci.* 2010, 3, 1311-1315; *Phys. Chem. Chem. Phys.* 2014, 16, 4720] proposed that the least-negative potential at which the pathway to each product becomes exergonic (downhill in free energy) is referred to as the limiting potential, for a single reduction pathway, the limiting potential is $\min(\Delta G(U = 0\text{ V})/e)$ of all elementary steps in the pathway. Meanwhile, Nørskov *et al.* also pointed that the free energy change of CO desorption ($*\text{CO} \rightarrow * + \text{CO}$) cannot participate in the calculation of limiting potential, because this is a nonelectrochemical step, increasing the potential does not

affect the free energy change [*J. Phys. Chem. Lett.* 2013, 4, 3, 388-392]. Therefore, our work and other reports [*Angew. Chem. Int. Ed.* 2021, 60, 1022-1032; *Angew. Chem. Int. Ed.* 2020, 59, 18572-185; *Adv. Mater.* 2019, 31, 1903470] calculated the limiting potentials according to the literatures mentioned above. It can be seen from Figure 5i and Table R3 that the limiting potentials of Fe-poN₄ and Fe-poN₄/Fe13 on ECR are -0.46 V and -0.02 V, respectively. However, the free energy changes for *COOH and *CO on Fe-pdN₄ are all downhill, so we chose a positive potential of 0.19 V as limiting potential for comparison with Fe-poN₄ and Fe-poN₄/Fe13. We have realized the limitations of such calculation method for limiting potential after the reviewer's kind reminder. The similar phenomenon can be observed in some high-impact literatures [*J. Phys. Chem. C* 2015, 119, 21345-21352; *Energy Environ. Sci.*, 2018, 11, 1204-1210; *ACS Catal.* 2019, 9, 10426-10439; *Chem* 2021, 7, 1-11], which investigated the competition between the ECR and the HER by comparing the free energy change of *COOH (ΔG_{*COOH}) and *H (ΔG_{*H}), and the more negative value of $\Delta G_{*COOH} - \Delta G_{*H}$ represents the higher CO selectivity. Thus, we have added Figure R11 as new Figure 5i and some new descriptions in the Revised Manuscript: “The Gibbs free energy diagram for HER was calculated and presented in Figure S22. The free energy change required for the *H formation of Fe-poN₄ (0.33 eV) is higher than that of Fe-pdN₄ (-0.36 eV) and Fe-poN₄/Fe13 (0.12 eV), indicating the key role of Fe-poN₄ sites in inhibiting HER and improving CO selectivity. Though Fe-poN₄/Fe13 shows an optimal HER energetics, a much lower value (-0.68 eV) of the difference between the free energy change of ΔG_{*COOH} and ΔG_{*H} than that of Fe-pdN₄ (0.17 eV) and Fe-poN₄ (0.13 eV) was achieved (Figure 5i), well explained the superior CO selectivity of Fe-poN-C/Fe shown in Figure. 3b.^{16,47,53}” (please see line 388-395 in the

Figure R11. The free energy for the formation of *COOH (ΔG_{*COOH}) or *H (ΔG_{*H}), and the difference between ΔG_{*COOH} and ΔG_{*H} .

Table R3. DFT calculated free energy change (ΔG , the free energy of the product minus that of the reactant) at 0 V or -0.3 V vs. RHE for the elementary steps of ECR to CO.

ΔG (eV)	Fe-pdN ₄		Fe-poN ₄		Fe-poN ₄ /Fe13	
	0 V	-0.3 V	0 V	-0.3 V	0 V	-0.3 V
* + CO ₂ + H ⁺ + e ⁻ → *COOH	-0.19	-0.49	0.46	0.16	-0.56	-0.86
*COOH + H ⁺ + e ⁻ → *CO + H ₂ O	-0.86	-1.16	-0.9	-1.2	0.02	-0.28
*CO → * + CO	1.16	1.16	0.55	0.55	0.65	0.55

Furthermore, our DFT calculations indeed predicted > 0.5 eV free energy required to desorb *CO from any of the three catalysts model using Perdew-Burke-Ernzerhof (PBE) functionals, such as 1.16 eV on Fe-pdN₄ model. Interestingly, Chan *et al.* predicted a weaker binding energy (0.42 eV) of *CO on Fe(-pd)N₄ model by using Heyd-Scuseria-Ernzerhof (HSE06) hybrid functionals and a strong binding energy (1.18 eV) of *CO on Fe(-pd)N₄ model by using PBE functionals [ACS Catal. 2020, 10, 7826-7835]. That is to say the *CO

adsorption energy is sensitive to the employed functionals in the DFT calculations. Nevertheless, the trend in the variation of the adsorption energies as a function of the compressive strain should not be altered by the employed functionals in the DFT calculations [Angew. Chem. Int. Ed. 2021, 60, 1022-1032]. In our study, the free energy required to desorb *CO on Fe-pdN₄ (1.16 eV) is much higher than that of Fe-poN₄ (0.55 eV) and Fe-poN₄/Fe₁₃ (0.65 eV), indicating that the Fe-pdN₄ sites with strong CO binding could result in a lower FE_{CO} than that of Fe-poN₄ sites and Fe-poN₄/Fe₁₃ sites, which is consistent with the experimental results.

Comment 5: While the authors focus on the CO₂ vibrational bands, I suggest them to look for the 250 cm⁻¹/330 cm⁻¹ and 1800cm⁻¹/2100 cm⁻¹ regions (CO vibrational bands). The relative intensities of these peak provide nice insights on the CO species (see J. Am. Chem. Soc. 2022, DOI: 10.1021/jacs.2c03172. <https://doi.org/10.1021/jacs.2c03172>; Angew. Chem. Int. Ed. 2021, 60, 16576–16584. <https://doi.org/10.1002/anie.202104114>; ACS Catal. 2021, 11, 7694–7701. <https://doi.org/10.1021/acscatal.1c01478>).

Response: Thank the reviewer for the warm comments. The above references provided by the reviewer show a lot of excellent works, which are of great significance in the field of *in situ* Raman. It would be very interesting to monitor the CO vibrational bands. We noticed that the references recommended by the reviewers detected the Raman peaks related to *CO intermediates, but we also found that the research systems of these references were all Cu-based catalysts (oxide-derived Cu or polycrystalline Cu). It is well known that Cu-based materials are the most promising candidate catalysts for producing C₂₊ products from the

electrochemical CO₂ reduction reaction, and the formation of *CO intermediates and further C-C coupling are necessary paths to generate C₂₊ products. Thus, the *CO intermediates are easily identified/monitored on Cu-based catalysts. However, no Raman peaks related to CO species were detected in our work. Considering that the electrocatalyst used in this work is M-N-C materials and the main product is CO, the transient concentration of the *CO intermediates may likely be too low to be observed. The *in situ* Raman results in our work are similar to those reported in the literature about M-N-C electrocatalysts for CO production [Angew. Chem. Int. Ed. 2021,60, 24022-24027; Adv. Funct. Mater. 2023, 2214609].

MINOR POINTS

Comment 1: Please mention that all the electric potential in the main text are given vs RHE.

To the best of my knowledge, this is never mentioned.

Response: We are very sorry for this negligence. As suggested by the reviewer, we have added the following new sentence in the revised supporting information: “All the electric potential potentials in this work were converted to versus reversible hydrogen electrode (vs. RHE) by the equation ($E_{\text{RHE}} = E_{\text{Ag/AgCl}} + 0.0591 \times \text{pH} + 0.210 \text{ V}$) unless otherwise specified.”

(please see Revised Supplementary Information, Page 5)

Comment 2: Few sentences/typos should be corrected, for instance “As a result, there is an urgent demand to excavate and screen an advanced electrocatalyst for energy-efficient ECR, (with the) core challenge (of reducing) overpotential [...]”; “An apparent decline in the FE_{CO} and j_{CO} of Fe-poN-C/Fe-(H₂SO₄/H₂O₂) w(as) observed (Figure. S16), (which) reveals that the Fe NPs possess a positive role”; “To deeply understand (how) the local structure, coordination

N type, Fe NPs and the oxygen species on carbon (contributes to) the performance of Fe-N-C catalysts on ECR to CO,”; “The (insight) is also confirmed by DFT calculations.

Response: Thank the reviewer very much for the warm and constructive comments.

According to the reviewer’s advice, we have corrected the following sentences/typos in the

Revised Manuscript: “... for energy-efficient ECR, with the core challenge of reducing overpotential while keeping high Faraday efficiency.” (please see line 38-39 in the Revised

Manuscript) “... Fe-poN-C/Fe-(H₂SO₄/H₂O₂) was observed (Figure. S17d,e), combined with the above two control experiments, which means that the Fe NPs possess a positive role....”

(please see line 270-271 in the Revised Manuscript) “To deeply understand how the local

structure, coordination N type, Fe NPs and the oxygen species on carbon, contributes to the

performance of Fe-N-C catalysts on ECR to CO.” (please see line 328-329) “...the insight is

also confirmed by DFT calculations.” (please see line 416 in the Revised Manuscript)

Furthermore, other incorrect expressions have also been modified in the Revised Manuscript

and marked as yellow colors.

Comment 3: Remove unformal English words, for instance “What’s more,”.

Response: We are grateful to the reviewer’s valuable comment. According to the reviewer’s suggestion, we have corrected the “What’s more” to “Furthermore” in the Revised Manuscript.

We checked the manuscript thoroughly and corrected other unformal English words in the

Revised Manuscript.

Reviewer #3 (Remarks to the Author):

In this work, Wang et al. reported a catalyst consisting of pyrrole type Fe-N₄ sites and Fe nanoparticles supported by less-oxygenated carbon matrix. Based on the experimental and theoretical studies, pyrrole type Fe-N₄ sites improved CO selectivity and Fe nanoparticles boosted the proton transfer. As a result, the catalyst showed a high CO FE above 99% under a low overpotential, and a high CEE over 80% maintain under a potential range of -0.12 to -0.6 V in a flow cell. However, I believe the conclusion cannot be fully supported by the authors' experimental and theoretical results. There are many concerns the authors need to clarify. Overall, I do not feel this manuscript meets the threshold for publication in Nature Communications.

Response: Thank the reviewer very much for reviewing our manuscript, and offering the sincere advices and constructive comments! They are all very important to improve the quality of this paper. We really appreciate it!

Comment 1: The type of coordination N in Fe-pdN-C(O) and Fe-poN-C(O) is speculated through precursor type, without experiment evidences (Line 140-143).

Response: We thank the reviewer for this insightful comment. Firstly, the mixture of FeTpp and carbon black can maintain the original pyrrole-type Fe-N₄ structure after a heat treatment at temperatures even up to 800 °C (Figure R12) according to the Veen's work [*J. Phys. Chem. B* 2002, 106, 12993-13001]. Likewise, the pyridine-type Fe-N₄ moiety can be reserved when the pyrolysis precursor is Fe-phenanthroline complex [*J. Am. Chem. Soc.* 2017, 139, 10790-10798; *Science* 2019, 364, 1091-1094]. Secondly, it can be seen from the FT-EXAFS spectra

(Figure 2b) that the position of the Fe-N peak in the Fe-poN-C(O) is close to that in FeTpp (with definite pyrrole-type Fe-N moiety), we infer that Fe-poN-C(O) mainly contains the pyrrole-type Fe-N structure (Fe-poN) and the Fe-pdN-C(O) may dominate the pyridine-type Fe-N (Fe-pdN) structure. Furthermore, as shown in the N 1s spectra (Figure 2e), the primary peaks assigned to Fe-N coordination in Fe-pdN-C(O) and Fe-poN-C(O) have a prominent difference at the energy position. Combined with the FT-EXAFS analysis and previous studies [*J. Electron Spectrosc. Relat. Phenom.* 1988, 46, 285-295; *Nat. Commun.* 2017, 8, 944; *Science* 2019, 364, 1091-1094], the main peak at around 398.7 eV from Fe-poN-C(O) can be ascribed to the Fe-poN structure, and the main peak at around 399.7 eV from Fe-pdN-C(O) can be ascribed to the Fe-pdN structure.

Figure R12. The schematic van Veen model of the structural change in a FeTpp molecule during heat treatment reproduced from Fig. 9 in *J. Phys. Chem. B* 2002, 106, 12993-13001.

We understand the reviewer's concern about the type of coordination N. Therefore, in order to further confirm the type of coordination N in Fe-pdN-C(O) and Fe-poN-C(O), according to the Hu's method [*Science* 2019, 364, 1091-1094], the N 1s spectra of two reference samples, Fe-phen complex with pyridine-type Fe-N₄ moiety and FeTpp with pyrrole-type Fe-N moiety, were chosen for comparison with that of Fe-pdN-C(O) and Fe-poN-C(O). As shown in Figure R13, the position of the Fe-N peak in the Fe-pdN-C(O) is close to that in Fe-phen and the

position of the Fe-N peak in the Fe-poN-C(O) is close to that in FeTpp, further demonstrating the types of coordination N in Fe-pdN-C(O) and Fe-poN-C(O) are mainly pyridine N and pyrrole N, respectively. We have added Figure R13 as Figure S6 and some new descriptions in the Revised Manuscript: “Moreover, the comparison of the N 1s spectra of Fe-pdN-C(O) and Fe-poN-C(O) with that of Fe-phen complex and FeTpp further confirmed that the types of coordination N in Fe-pdN-C(O) and Fe-poN-C(O) are mainly pyridine N and pyrrole N, respectively (Figure S6).¹⁸” (please see line 181-184 in the Revised Manuscript)

Figure R13. N 1s XPS spectra of Fe-pdN-C(O), Fe-poN-C(O), Fe-phen complex and FeTpp.

Comment 2: The high-resolution N 1s spectra should be fitted by different N species. Additionally, the Fe-N bond length in Fe-poN-C/Fe is shorter than that in Fe-poN-C(O), the authors think that derive from the strong interaction between Fe NPs and Fe-N₄ sites, while the Fe-N binding energy in N 1s spectra of Fe-poN-C/Fe and Fe-poN-C(O) is same, which seems contradictory.

Response: We thank the reviewer’s constructive suggestion about the N 1s spectra. According to the reviewer’s suggestion, the high-resolution N 1s spectra have been fitted by different N species (Figure R14, as Figure 2e in the Revised Manuscript). Indeed, the main peaks

assigned to Fe-N coordination in N 1s spectra of Fe-poN-C/Fe and Fe-poN-C(O) is same, while the Fe-N bond length in WT-XAFS spectra of Fe-poN-C/Fe is shorter than that of Fe-poN-C(O). Similar phenomena have also been observed in the published literature on the coexistence of metal NPs and M-N_x sites [*Angew. Chem. Int. Ed.* 2022, 61, e202203335; *Adv. Mater.* 2022, 34, 2107291; *Adv. Sci.* 2021, 8, 2001881]. Therefore, the Fe-N bond length in Fe-poN-C/Fe is shorter than that in Fe-poN-C(O) may derive from the strong interaction between Fe NPs and Fe-N_x groups.^{26,27} We understand the reviewer's concern about the Fe-N analysis from the results of XPS and XAFS. It should be noted that the Fe-N analysis obtained from XPS measurements comes from the N 1s spectra (the test element is N), while the Fe-N analysis obtained from XAFS testing comes from the Fe K-edge spectra (the test element is Fe).

Figure R14. N 1s spectra of Fe-pdN-C(O), Fe-poN-C(O) and Fe-poN-C/Fe.

Comment 3: All electrochemistry figures lack error bars.

Response: Thanks for the valuable suggestion. Error bars have been added into all electrochemistry figures containing the Faradic efficiency and current density.

Comment 4: In SCN⁻ poison experiments, why SCN⁻ cannot poison Fe nanoparticles (Lines

240-242)? What is the mechanism of SCN^- poison?

Response: Thank the reviewer for the good questions. The lone pair electrons on heteroatom of SCN^- can bind to Fe species with empty orbitals in the form of coordination bonds. Wang *et al.* have proven that SCN^- as a poisoning reagent could block the active Fe-N_x sites in Fe-N-C catalysts [*J. Am. Chem. Soc.* 2014, 136, 10882-10885]. Therefore, to identify whether atomic Fe-N_x sites are the intrinsically active sites in the multicomponent catalysts, numerous literatures performed SCN^- poisoning experiments [*J. Am. Chem. Soc.* 2016, 138, 3570-3578; *J. Am. Chem. Soc.* 2017, 139, 10790-10798; *Adv. Mater.* 2022, 34, 2107291]. In theory, SCN^- can bind to Fe-N_x sites and Fe nanoparticles because the Fe atoms in them contain empty orbitals. However, previous reports have demonstrated that SCN^- ions are only bind to Fe-N_x sites but not to Fe nanoparticles [*ACS Catal.* 2022, 12, 7517-7523; *ACS Catal.* 2022, 12, 5595-5604], so SCN^- cannot poison Fe nanoparticles.

Comment 5: The authors treated Fe-poN-C/Fe catalyst with 0.5 M H_2SO_4 containing certain 30% H_2O_2 to remove Fe NPs and increase O species content simultaneously. It is not sufficient to confirm the role of Fe NPs and O species, due to the presence of two variables (Lines 246-255).

Response: Thank the reviewer for the warm comments. We agree with the reviewer's point that it is not sufficient to confirm the role of Fe NPs and O species using only one control experiment. Therefore, the control experiments for removing Fe nanoparticles and increasing oxygen content have been conducted separately. In order to confirm the role of oxygen species on the ECR performance, Figure R15 present the full XPS spectra, FE_{CO} and j_{CO} of

Fe-poN-C(O) before and after H₂O₂/H₂SO₄ treatment. Likewise, Figure R16 present the TEM spectra, FE_{CO} and j_{CO} of Fe-poN-C/Fe before and after H₂SO₄ treatment to confirm the role of Fe NPs. The results and analysis of the new experiments have added to the Revised Manuscript: “The effect of Fe NPs and oxygen species on the ECR performance was further studied by control experiments. Firstly, Fe-poN-C(O) catalyst was treated with H₂O₂/H₂SO₄ solutions (0.5 M H₂SO₄ containing certain 30% H₂O₂) under 80 °C for 24 h to increase the content of oxygen species. XPS results exhibit that more oxygen species have been introduced on Fe-poN-C(O) after H₂O₂/H₂SO₄ treatment (Figure S15a). The FE_{CO} and j_{CO} of the treated Fe-poN-C(O) (named as Fe-poN-C(O)-(H₂O₂/H₂SO₄)) decreased significantly as compared with Fe-poN-C(O) (Figure S15b,c), indicating the oxygen species on carbon supports are unfavorable for promoting ECR. Secondly, Fe-poN-C/Fe catalyst was treated with H₂SO₄ solutions (0.5 M H₂SO₄) under 80 °C for 24 h to remove the Fe NPs. The HAADF-STEM images of the treated Fe-poN-C/Fe (named as Fe-poN-C/Fe-(H₂SO₄)) show that the Fe NPs are almost all removed and single-atom Fe sites remained (Figure S16a,b). Performance tests exhibits the FE_{CO} at low potentials and j_{CO} decreased after the removal of Fe NPs (Figure S16c,d), which indicates that the existence of Fe NPs is benefit for reducing overpotential on ECR. Furthermore, the HAADF-STEM images and XPS results shown in Figure S17a-c reveal the Fe NPs in Fe-poN-C/Fe are removed while more oxygen species are introduced after H₂O₂/H₂SO₄ treatment. An apparent decline in the FE_{CO} and j_{CO} of Fe-poN-C/Fe-(H₂SO₄/H₂O₂) was observed (Figure. S17d,e), combined with the above two control experiments, which means that the Fe NPs possess a positive role in reducing overpotential and oxygen species exhibits a negative role in improving CO selectivity.” (please see line

Figure R15. (a) Full XPS spectra, (b) FE_{co} and (c) j_{co} of Fe-pdN-C(O) before and after H₂O₂/H₂SO₄ treatment.

Figure R16. (a) HAADF-TEM and (b) aberration-corrected HAADF-STEM of Fe-poN-C/Fe after H₂SO₄ treatment. (c) FE_{CO} and (d) j_{CO} of Fe-poN-C/Fe before and after H₂SO₄ treatment.

Comment 6: The 0.5 M KHCO₃ electrolyte was used in H type cell, why the electrolyte in flow cell was changed to 1 M KOH aqueous (Line 264)?

Response: We thank the reviewer for this good question. The electrolyte plays a critical role in assisting a catalyst to achieve its latent capability. Thus, the selection of electrolyte in ECR is important. KHCO₃ is the most frequently used electrolyte for aqueous ECR, especially in H type cell [*J. Am. Chem. Soc.* 2017, 139, 14889-14892; *Nat. Commun.* 2021, 12, 1449]. The adsorbed K⁺ ion was theoretically demonstrated to possess the effect on activation of CO₂ through the local electric field [*Nature* 2016, 537, 382-386; *Energy Environ. Sci.* 2019, 12, 3001]. HCO₃⁻ was supposed to be able to rapidly equilibrate with dissolved CO₂, which increased the concentration of dissolved CO₂ near the electrode surface. However, the current

density of CO₂ conversion in H type cell would be largely limited by the low solubility and slow diffusion of CO₂ in aqueous systems, and hardly satisfy the industrial-relevant current density (>100 mA·cm⁻²) [*Science* 2018, 360, 783-787; *Nat. Commun.* 2020, 11, 593]. To solve these problems, the flow cell using gas-diffusion electrodes (GDEs) have been widely studied in recent years. GDEs enable the direct feed of gas-phase CO₂ to the cathode, which is important for overcoming mass-transport limitations and achieving high current density [*Nat. Energy* 2022, 7, 130-143; *Energ. Environ. Sci.* 2019, 12, 1442-1453; *Nat. Commun.* 2023, 14, 2062]. Meanwhile, KOH is popularly used as electrolyte in ECR reaction under flow cell conditions [*Nat. Commun.* 2021, 12, 1449; *Adv. Mater.* 2022, 2205262; *Angew. Chem. Int. Ed.* 2022, e202206233; *Energy Environ. Sci.*, 2021,14, 2349-2356], because it possesses low solution resistance and the ability to inhibit hydrogen evolution side reaction [*Chem*, 2018, 4, 2571-2586; *Appl. Catal. B-environ* 2021, 288, 120003], which can further improve the current density of the target product. Thus, in this study, the high-current activity of Fe-poN-C/Fe was evaluated in a flow cell device equipped with gas diffusion electrode using 1 M KOH as electrolyte.

Comment 7: The stability test was performed in H type cell or flow cell (Lines 285-286)? If it was in flow cell, the current density is too low.

Response: We thank the reviewer for this pertinent question. The stability test of Fe-poN-C/Fe was performed in a flow cell. We agree that the current density for stability test is low. In fact, the durability of Fe-N-C type catalysts under high current density has always been a challenge in this field, the high current density is difficult to maintain and the CO Faraday

efficiency will continue to decline during long-term stability test in a flow cell. For example, Gu *et al.* reported that Fe³⁺-N-C catalysts exhibit a current density about 50 mA·cm⁻² during a stability test of 28 h at a constant potential of -0.41 V (with *iR*) while the FE_{CO} decreased from ~95% to <90% [*Science* 2019, 364, 1091-1094], and the Liu group reported that the FE_{CO} of Fe₁-NSC catalyst decreased from ~98% to ~90% under a constant current density of 40 mA·cm⁻² with a potential about -0.45 V (with *iR*) during a stability test only 20000 s [*Angew. Chem. Int. Ed.* 2022, 61, e202206233]. It should be noted that Fe³⁺-N-C and Fe₁-NSC represent the almost most state-of-the-art Fe-N-C catalysts for ECR to CO. In our study, stability tests over 100 h show that Fe-poN-C/Fe catalysts can maintain nearly 100% FE_{CO} with a current density about 50 mA·cm⁻² (>40 mA·cm⁻²) at a constant potential of -0.3 V (without *iR*). Furthermore, the loading of Fe-poN-C/Fe catalyst is only 1 mg·cm⁻², while that of Fe³⁺-N-C and Fe₁-NSC are both up to 2.5 mg·cm⁻². Thus, we have significantly improved the long-term durability of Fe-N-C catalyst, including current density and FE_{CO}, even with low catalysts loading at lower potential.

Comment 8: The figure numbers of Figure 5 were wrong in the manuscript. Line 319, the authors think the RDS on Fe-poN₄ is the *COOH formation, however, it is obvious that the RDS on Fe-poN₄ in Figure 5e is *CO desorption. Please show the free energy change value.

Response: We are very sorry for this negligence (the wrong figure numbers). We have corrected the “Figure. 4d”, “Figure. 4f” and “Figure. 4g-h” to “Figure. 5d”, “Figure. 5f” and “Figure. 5g-h” in the Revised Manuscript. In addition, we admit that the expression of “the RDS on Fe-poN₄ is the *COOH formation” is incorrect according to the reviewer’s advice

and Table R4. We have corrected the wrong expression and added the new descriptions in the Revised Manuscript: “Although Fe-poN₄ exhibits lower free energy change (0.55 eV) for *CO desorption than that of Fe-pdN₄, its free energy change (0.16 eV) required for *COOH formation obviously increased.” (please see line 338-340 in the Revised Manuscript) We thank the reviewer again for this kind comment.

Table R4. DFT calculated free energy change (ΔG , the free energy of the product minus that of the reactant) at 0 V or -0.3 V vs. RHE for the elementary steps of ECR to CO.

Reaction step	Fe-pdN ₄		Fe-poN ₄		Fe-poN ₄ /Fe13	
	0 V	-0.3 V	0 V	-0.3 V	0 V	-0.3 V
* + CO ₂ + H ⁺ + e ⁻ → *COOH	-0.19	-0.49	0.46	0.16	-0.56	-0.86
*COOH + H ⁺ + e ⁻ → *CO + H ₂ O	-0.86	-1.16	-0.9	-1.2	0.02	-0.28
*CO → * + CO	1.16	1.16	0.55	0.55	0.65	0.55

REVIEWERS' COMMENTS

Reviewer #1 (Remarks to the Author):

The authors have addressed some of my comments, but I am still not convinced by the reply to my previous comments.

1. Again, it is important to indicate at which current density high energy efficiency (CEE of 97.1%, in this work) is obtained, as such a high value of 97.1% is usually impossible under considerable current densities. Reporting corresponding current densities (at which high selectivity and/or energy efficiency is obtained) is a common practice. And, a low overpotential does not necessarily mean high performance.

2. In the original manuscript, the authors stated "Fe-poN-C/Fe catalyst was treated with 0.5 M H₂SO₄ containing certain 30% H₂O₂ under 80 °C for 24 h to remove Fe NPs and increase the content of oxygen species."

In the revised manuscript, the authors stated "Fe-poN-C(O) catalyst was treated with H₂O₂/H₂SO₄ solutions (0.5 M H₂SO₄ containing certain 30% H₂O₂) under 80 °C for 24h to increase the content of oxygen species."

This is contradicted! Why cannot H₂O₂/H₂SO₄ remove Fe NPs in the revised version?

Reviewer #2 (Remarks to the Author):

I sincerely thank Wang et al. I've been reviewing tens of paper and it is the first time that I see such a open approach to consider both mine and other reviewers' comments. When possible, all my major points are now tackled in the revised version and I have significantly learnt through the discussion with the authors. I personally think the work is now ready for publication in Nat. Commun., provided that the authors further check few minor issues.

(1) Although the authors have now mentioned Nørskov's work in the Supplementary files, no mention is given in the main text. I think the authors should explicitly mention that they use the Computational Hydrogen Electrode (with relative references) also within the main text. Please note that the first application of CHE was reported in J. Phys. Chem. B 2004, 108, 17886–17892.

<https://doi.org/10.1021/jp047349j>, which should then be cited both in main text and SI.

(2) To make the Data Availability statement real, I suggest the authors to upload all the DFT datasets to open source repositories such as ioChem-BD (<https://www.iochem-bd.org/>). They can request an account, upload the simulations and set up an embargo period until the work is not yet published. Such database has been suggested by Nature itself.

(3) Fig. S24. Please double-check *H₂O dissociation energy for clean carbon. I think that a ΔG of 3 eV is unrealistic. Probably some issues with the reference energies (energy of C system) occur.

(4) Fig.5i. Please change the label in the panel from *COOH - *H to G_{_{*}COOH}- G_{_{*}H}, since the operation is applied to Gibb free energy rather than the reactant species.

Reviewer #3 (Remarks to the Author):

The authors have addressed the reviewers' comments, who now recommends the publication in Nature Communications.

Response to Reviewers

Reviewer #1 (Remarks to the Author):

The authors have addressed some of my comments, but I am still not convinced by the reply to my previous comments.

Comment 1: Again, it is important to indicate at which current density high energy efficiency (CEE of 97.1%, in this work) is obtained, as such a high value of 97.1% is usually impossible under considerable current densities. Reporting corresponding current densities (at which high selectivity and/or energy efficiency is obtained) is a common practice. And, a low overpotential does not necessarily mean high performance.

Response: Thank the reviewer for the constructive comments. In our work, Fe-poN-C/Fe exhibits a high CEE of 97.1% with a current density of $-14.1 \text{ mA}\cdot\text{cm}^{-2}$. To clearly clarify this point in this study, we have added some new descriptions in the Revised Manuscript: “...which also achieves an ultrahigh cathode energy efficiency (CEE) of 97.1% with nearly 100% FE_{CO} and a current density of $-14.1 \text{ mA}\cdot\text{cm}^{-2}$ at an ultralow overpotential of 21 mV in a flow cell...” (please see line 76 in the Revised Manuscript) “...and the maximum CEE of 97.1% with a current density of $-14.1 \text{ mA}\cdot\text{cm}^{-2}$ was obtained at an ultralow overpotential of 21 mV...” (please see line 304 in the Revised Manuscript) “...and a maximum CEE of 97.1% with nearly 100% FE_{CO} and a current density of $-14.1 \text{ mA}\cdot\text{cm}^{-2}$ was obtained at an ultralow overpotential of 21 mV...” (please see line 416 in the Revised Manuscript)

Indeed, a low overpotential does not necessarily mean high performance, so we have decided to remove the word of “highly energy-efficient” in the title.

Comment 2: In the original manuscript, the authors stated “Fe-poN-C/Fe catalyst was treated with 0.5 M H₂SO₄ containing certain 30% H₂O₂ under 80 °C for 24 h to remove Fe NPs and increase the content of oxygen species.”

In the revised manuscript, the authors stated “Fe-poN-C(O) catalyst was treated with H₂O₂/H₂SO₄ solutions (0.5 M H₂SO₄ containing certain 30% H₂O₂) under 80 °C for 24h to increase the content of oxygen species.”

This is contradicted! Why cannot H₂O₂/H₂SO₄ remove Fe NPs in the revised version?

Response: We understand the reviewer's misconception about the H₂O₂/H₂SO₄ treatment. First, the carbon materials can be oxidized by using H₂O₂ as an oxidizing agent in H₂SO₄ solutions based on the modified Hummers method [*Energy Environ. Eng.* 2014, 2, 58-63; *Nat. Mater.* 2020, 19, 436-442]. Second, the treatment of H₂O₂/H₂SO₄ can not only increase oxygen species but also remove metal nanoparticles due to the presence of H₂SO₄ in the solution. It should be noted that Fe-poN-C/Fe catalyst consists of Fe NPs, Fe-N₄ sites and carbon support, while Fe-poN-C(O) catalyst only consists of Fe-N₄ sites and carbon support. Therefore, the H₂O₂/H₂SO₄ treatment on Fe poN-C (O) catalyst can only increase the content of oxygen species on the carbon support, while the H₂O₂/H₂SO₄ treatment on Fe poN-C (O) catalyst can not only increase oxygen species on the carbon support but also remove Fe NPs in the catalysts.

Reviewer #2 (Remarks to the Author):

I sincerely thank Wang et al. I've been reviewing tens of paper and it is the first time that I see such a open approach to consider both mine and other reviewers' comments. When possible, all my major points are now tackled in the revised version and I have significantly learnt through the discussion with the authors. I personally think the work is now ready for publication in Nat. Commun., provided that the authors further check few minor issues.

Comment 1: Although the authors have now mentioned Nørskov's work in the Supplementary files, no mention is given in the main text. I think the authors should explicitly mention that they use the Computational Hydrogen Electrode (with relative references) also within the main text. Please note that the first application of CHE was reported in *J. Phys. Chem. B* 2004, 108, 17886–17892. <https://doi.org/10.1021/jp047349j>, which should then be cited both in main text and SI.

Response: Thank the reviewer for the warm comments. We have cited the suggested literatures [*J. Phys. Chem. B* 2004, 108, 17886-17892; *Energy Environ. Sci.* 2010, 3, 1311-1315] as ref. 60 and 61 in the main manuscript.

Comment 2: To make the Data Availability statement real, i suggest the authors to upload all the DFT datasets to open source repositories such as ioChem-BD (<https://www.iochem-bd.org/>). They can request an account, upload the simulations and set up an embargo period until the work is not yet published. Such database has been suggested by Nature itself.

Response: Thank the reviewer for the constructive suggestion. In order to facilitate future readers to directly obtain DFT calculation data and other data, such as the XRD spectra, XAS spectra, XPS spectra, product quantification, *in situ* Raman spectra, Source Data for the above analyses have been deposited in the Figshare database, this database is suggested by Nature communications. Furthermore, Supplementary Data as a .zip file containing the crystallographic information files of DFT calculation results have also been deposited in the Figshare database. We have uploaded the file of Source Data and Supplementary Data, and related introductions have been added in the section of **Data availability**: “The main data supporting the findings of this study are available within the article and its Supplementary Information or are available from the corresponding authors upon reasonable request. Source data for Figures 1-5 and Supplementary Data containing the crystallographic information files of DFT calculation results have been deposited in the Figshare database under accession code (<https://doi.org/10.6084/m9.figshare.23713836>).” (please see line 547-553 in the *Revised Manuscript*)

Comment 3: Fig. S24. Please double-check *H₂O dissociation energy for clean carbon. I think that a ΔG of 3 eV is unrealistic. Probably some issues with the reference energies (energy of C system) occur.

Response: According to the reviewer's advice, we have double-checked the free energy of C model with H₂O species (*H₂O, -680.6222 eV) and C model with H-OH species (*H-OH, -677.8641 eV), and a *H₂O dissociation energy (2.76 eV) for C model using

in this work was definitely obtained.

Comment 4: Fig.5i. Please change the label in the panel from $*\text{COOH} - *H$ to $G*\text{COOH}-G*H$, since the operation is applied to Gibb free energy rather than the reactant species.

Response: Thanks for the valuable suggestion. We have corrected “ $*\text{COOH}$ ”, “ $*H$ ” and “ $*\text{COOH} - *H$ ” to “ $G*\text{COOH}$ ”, “ $G*H$ ” and “ $G*\text{COOH} - G*H$ ” in the revised Figure 5i.

Figure 5i. The free energy for the formation of $*\text{COOH}$ ($G*\text{COOH}$) or $*H$ ($G*H$), and the difference between $G*\text{COOH}$ and $G*H$.

Reviewer #3 (Remarks to the Author):

The authors have addressed the reviewers' comments, who now recommends the publication in Nature Communications.

Response: We sincerely appreciate the reviewer's comments of our study.